# SILO LANGUAGE MODELS: ISOLATING LEGAL RISK IN A NONPARAMETRIC DATASTORE

**Sewon Min**[*1]    **Suchin Gururangan**[*1]    **Eric Wallace**[2]    **Weijia Shi**[1]
**Hannaneh Hajishirzi**[1,3]    **Noah A. Smith**[1,3]    **Luke Zettlemoyer**[1]
[1]University of Washington    [2]UC Berkeley    [3]Allen Institute for AI
{sewon,sg01,hannaneh,nasmith,lsz}@cs.washington.edu    ericwallace@berkeley.edu

## ABSTRACT

The legality of training language models (LMs) on copyrighted or otherwise restricted data is under intense debate. However, as we show, model performance significantly degrades if trained only on low-risk text (e.g., out-of-copyright books or government documents), due to its limited size and domain coverage. We present SILO, a new language model that manages this risk-performance tradeoff during inference. SILO is built by (1) training a parametric LM on the OPEN LICENSE CORPUS (OLC), a new corpus we curate with 228B tokens of public domain and permissively licensed text and (2) augmenting it with a more general and easily modifiable nonparametric datastore (e.g., containing copyrighted books or news) that is only queried during inference. The datastore allows use of high-risk data without training on it, supports sentence-level data attribution, and enables data producers to opt out from the model by removing content from the store. These capabilities can foster compliance with data-use regulations such as the *fair use* doctrine in the United States and the GDPR in the European Union. Our experiments show that the parametric LM struggles on its own with domains not covered by OLC. However, access to the datastore greatly improves out of domain performance, closing 90% of the performance gap with an LM trained on the Pile, a more diverse corpus with mostly high-risk text. We also analyze which nonparametric approach works best, where the remaining errors lie, and how performance scales with datastore size. Our results suggest that it is possible to build high quality language models while mitigating legal risk.

## 1   INTRODUCTION

Large language models (LMs) are under widespread legal scrutiny, in large part because they are trained on copyrighted content, which may infringe on the rights of data producers (Metz, 2022; Vincent, 2023; J.L. et al. v. Alphabet Inc., 2023; Brittain, 2023). At the heart of this discussion is the inherent tradeoff between legal risk and model performance. Training only on data sources such as public domain, non-copyrightable, or otherwise permissively licensed data significantly degrades performance (as we show in §4.1). This limitation arises from the scarcity of permissive data and its narrow specificity to sources which are largely different from common LM corpora that cover more diverse domains (Raffel et al., 2020; Gao et al., 2020; Together, 2023).

In this paper, we demonstrate that it is possible to improve the risk-performance tradeoff by segregating training data into two distinct parts of the model: parametric and nonparametric (Figure 1). We learn LM parameters on *lower-risk* data (i.e., data under the most permissive licenses), and then use *higher-risk* data (i.e., data under copyright, restrictive licenses, or unknown licenses) in an inference-time-only nonparametric component (called a *datastore*). With nonparametric datastores, we can *retrieve* higher-risk data to improve model predictions without training on it. The datastore can be easily updated at any time, and allows creators to remove their data from the model entirely, at the level of individual examples. This approach also attributes model predictions at the sentence-level, enabling credit assignment to data owners. These new capabilities enable better alignment of the model with various data-use regulations, e.g., the *fair use* doctrine in the United States (Henderson

---

[*]Equal Contribution.

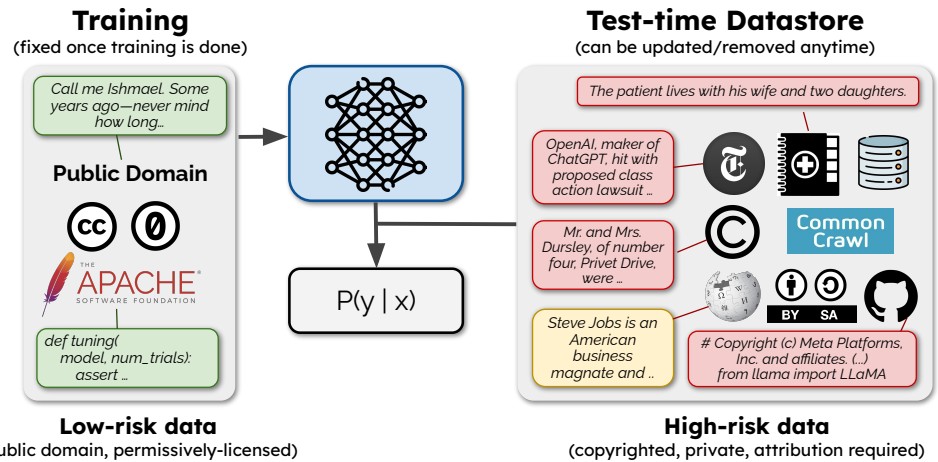

Figure 1: **An overview of SILO.** We train a *parametric* language model on low-risk datasets that contain public domain text (e.g., copyright-expired books) and permissively licensed code. At inference time, we use a *nonparametric datastore* that can include high-risk data, including medical text with personally-identifiable information, copyrighted news, copyrighted books, data requiring attribution, and code under non-permissive licenses (counterclockwise from the top of figure). The datastore can be modified at any time, e.g., to respond to opt-out requests.

et al., 2023) and the GDPR in the European Union (Zhang et al., 2023), as detailed in §A. This is in contrast to parametric models, where removing high-risk data is infeasible after training (Bourtoule et al., 2020; Carlini et al., 2021) and data attribution is difficult (Zhang et al., 2021; Han et al., 2023).

We introduce SILO, a new language model that follows our proposal (§3). The parametric component in SILO is trained on a new pretraining corpus, the **OPEN LICENSE CORPUS** (**OLC**, §3.1), which we curate to include data under three types of permissive licenses, from public domain to Creative Commons. OLC is diverse but has a domain distribution that is very different from typical pretraining corpora: it is dominated by code and government text. This leads to a new challenge of generalization beyond highly limited training domains, which we call *extreme domain generalization*. We train three 1.3B-parameter LMs on varying subsets of OLC, and then construct a test-time datastore that can include high-risk data, employing a retrieval method to make use of the datastore's contents during inference. We compare two widely studied retrieval methods: a nearest-neighbors approach ($k$NN-LM) that uses a nonparametric next-token prediction function (Khandelwal et al., 2020) and a retrieval-in-context approach (RIC-LM) that retrieves text blocks and feeds them to the parametric LM in context (Shi et al., 2023; Ram et al., 2023).

We evaluate SILO in language modeling perplexity on 14 different domains, covering both in-domain and out-of-domain data with respect to OLC (§4). These domains highlight specific legal risks, e.g., copyrighted materials such as books, news and user reviews, or private data such as emails and clinical notes. We also evaluate models in zero-shot downstream performance on ten text classification datasets. We compare SILO to Pythia (Biderman et al., 2023), a parametric LM with a similar parameter count trained on less restricted data with much wider domain coverage (Gao et al., 2020).[1] We first show that parametric-only SILO is competitive on domains covered by OLC but falls short out-of-domain, confirming the challenge of extreme domain generalization. However, adding an inference-time datastore to SILO effectively addresses this challenge. Comparing the two methods of retrieving over this datastore, we find that while both $k$NN-LM and RIC-LM significantly improve out-of-domain performance, the former generalizes better than the latter, allowing SILO to reduce the gap with the Pythia baseline by 90% on average across all domains. Further analysis attributes these improvements to two factors: (1) $k$NN-LM strongly benefits from scaling the datastore and (2) the nonparametric next-token prediction in $k$NN-LM is robust to domain shift. Altogether, our study suggests important future directions such as closing the remaining gaps by scaling the datastore size and improving the runtime speed of SILO.

---

[1]The Pile contains a large amount of copyrighted or restrictively licensed data, e.g., most content in its Books3, ArXiv, Github, OpenWebText, YoutubeSubtitles, and Common Crawl subsets.

## 2 BACKGROUND & RELATED WORK

State-of-the-art LMs are trained on text corpora with trillions of tokens (Brown et al., 2020; Raffel et al., 2020; Gao et al., 2020; Together, 2023). These training sets are built by combining (1) manually selected sources such as Wikipedia, book collections, and GitHub and (2) web pages collected through web-crawling services such as Common Crawl. Such data includes sources such as GitHub repositories and book collections that contain text with highly restrictive licenses (Bandy & Vincent, 2021). This approach has led to intense ongoing debate on the legality of language models; we provide legal background in §A.

The goal of our work is not to weigh in on legal discussions; instead, we study the feasibility of developing technologies that explicitly manage legal risk. SILO places all copyrighted data in a nonparametric datastore and supports instance-level attribution and data opt-out, which can increase the likelihood of a successful fair use defense (Henderson et al., 2022) and adhering to the General Data Protection Regulation (GDPR). Attribution and opt-out are fundamental features (§3.2), in contrast to other techniques like post-hoc training data attribution (Koh & Liang, 2017; Han et al., 2023) and the removal of the effect of particular training examples from parameters (Cao & Yang, 2015; Jang et al., 2023b) which lack inherent guarantees and are hard to scale.

**Prior work in copyright risk mitigation.** The most straightforward approach to avoid copyright infringement is to filter training data to only include permissive licenses. This has been done in prior work, primarily for code-based datasets (e.g., Kocetkov et al., 2023; Fried et al., 2023; Together, 2023) and scientific text (e.g., Soldaini & Lo, 2023). Extending a similar approach to a wider range of domains remains unclear, because permissive data is extremely scarce in most domains, e.g., books and news. For the same reason, Henderson et al. (2023) has suggested that restricting the training data to public domain or otherwise permissively licensed data may be impractical. In this work, we show that there is in fact a large number of tokens from data sources with permissive licenses, but the key challenge instead arises from the highly skewed domain distribution.

## 3 SILO

We introduce SILO, an LM that takes a prefix of text $x$ and outputs a next-word probability distribution over the vocabulary $P(y \mid x)$. SILO combines a parametric LM trained on permissive data (§3.1) with a nonparametric datastore based on less restricted data (§3.2). The key idea is to use low-risk data to estimate model parameters, and to use high-risk data only in a nonparametric datastore. This is based on the motivation that model parameters should be learned conservatively, since training data is difficult to remove or trace after model training is completed. In contrast, a nonparametric datastore offers greater flexibility, as it can be easily updated, grown, or filtered, supports data opt-out at the level of individual examples, and provides attributions for free to every model prediction.

### 3.1 PARAMETRIC COMPONENT & OPEN LICENSE CORPUS (OLC)

For the parametric component of SILO, we use a 1.3B decoder-only transformer LM (Vaswani et al., 2017) based on the LLaMA architecture (Touvron et al., 2023) as implemented in OpenLM.[2] This model uses a fixed set of parameters at both training and inference time.

The parametric component of SILO should be learned on the *lower-risk* data, i.e., data in the public domain or under permissive licenses. To this end, we introduce the **OPEN LICENSE CORPUS (OLC)**, a large collection of permissive textual datasets across multiple domains, comprising 228B tokens. It is essential to note that the permissivity of a license type is not a binary decision but rather a spectrum. Therefore, we establish three levels of permissive licenses: **Public domain** (PD), **Permissively licensed software** (SW) and **Attribution licenses** (BY). This allows model developers to select the boundary that aligns with their preferences. See §B.1 for the description of each license type. We train the models on varying subsets of licenses—from PD and PDSW to PDBYSW—to accommodate different risk tolerances. Later sections demonstrate our empirical findings are consistent across all possible boundaries (§4, §D.2, Figure 3).

The data generally falls into eight different domains:

---

[2]https://github.com/mlfoundations/open_lm

Table 1: **OLC is large but its distribution is different from that of typical pretraining corpora like the Pile.** Data distribution of OLC (PD, PDSW, PDSWBY) in comparison to the Pile (Gao et al., 2020), a common LM training dataset that is not specifically designed for legal permissibility. We report the number of tokens in billions, and the relative frequency. †: There is no explicit news domain in the Pile, but news sites are found to be some of the most representative data sources in Common Crawl (Dodge et al., 2021).

| Domain | PD | | PDSW | | PDSWBY | | *The Pile* | |
|---|---|---|---|---|---|---|---|---|
| | Tokens (B) | % | Tokens (B) | % | Tokens (B) | % | Tokens (B) | % |
| Code | 0.0 | 0.0 | 58.9 | 59.1 | 58.9 | 25.8 | 32.6 | 9.8 |
| Legal | 27.1 | 86.2 | 27.1 | 27.2 | 27.2 | 11.9 | 30.8 | 9.3 |
| Conversation | 0.0 | 0.0 | 5.9 | 5.9 | 27.2 | 11.9 | 33.1 | 10.0 |
| Math | 0.0 | 0.0 | 3.5 | 3.5 | 3.5 | 1.50 | 7.1 | 2.1 |
| Books | 2.9 | 9.3 | 2.9 | 2.9 | 2.9 | 1.3 | 47.1 | 14.2 |
| Science | 1.2 | 3.8 | 1.2 | 1.2 | 71.5 | 31.3 | 86.0 | 26.0 |
| News | 0.2 | 0.7 | 0.2 | 0.2 | 0.2 | 0.1 | -† | -† |
| Wikipedia | 0.0 | 0.0 | 0.0 | 0.0 | 37.0 | 16.2 | 12.1 | 3.7 |
| Unverified web | 0.0 | 0.0 | 0.0 | 0.0 | 0.0 | 0.0 | 83.1 | 25.0 |
| Total | 31.4 | 100.0 | 99.6 | 100.0 | 228.3 | 100.0 | 331.9 | 100.0 |

- PD BY **Legal:** We curate legal text from the Pile of Law (Henderson et al., 2022), an amalgation of 31 different sources of text related to civil court cases, patents, and other legal and governmental works, either licensed as public domain or CC-BY. We also gather public domain text from the Case Law Access Project (Caselaw Access Project), which covers over 6.5 million decisions published by state and federal courts throughout U.S. history.

- SW **Code:** We use the Github subset of the RedPajama dataset (Together, 2023), which contains code from Github repositories with three permissive software licenses: MIT, Apache, and BSD.

- SW BY **Conversation:** We source conversational text under permissive software licenses from the HackerNews (MIT license) and the Ubuntu IRC (Apache license) subsets of the Pile (Gao et al., 2020). We also use the Stackexchange subset of the RedPajama dataset (Together, 2023) and a Stackoverflow corpus from Kaggle,[3] both under the CC-BY-SA license.

- SW **Math:** We source mathematical text from the Deepmind Mathematics (Saxton et al., 2019) and the AMPS (Hendrycks et al., 2021) datasets, both of which are under the Apache license.

- PD BY **Science:** We source scientific text from ArXiv abstracts that are in the public domain (ArXiv, 2023). We also collect full-text articles from the Semantic Scholar Research Corpus (Lo et al., 2020, S2ORC), either licensed as public domain or CC-BY.

- PD **Books:** We source books from the Gutenberg corpus (Project Gutenberg), which are copyright-expired books that are in the public domain.

- PD BY **News:** We collect public domain news text from the English subset of the MOT corpus (Palen-Michel et al., 2022). We also collect text from Wikinews, which is under CC BY-SA.

- BY **Encyclopedic:** Finally, we include a large set of Wikipedia from the subset included in RedPajama (Together, 2023). We follow RedPajama in using Wikipedia snapshots from 20 languages even though the model primarily focuses on English.

Following Kandpal et al. (2022) and Lee et al. (2022), we deduplicate text using a document-level filter that considers $n$-gram overlap (Groeneveld, 2023). We first deduplicate within each domain to remove redundant documents from similar sources (e.g. Case Law and the Pile of Law), and then perform deduplication against the validation and test datasets of the Pile to avoid test leakage.

**Data statistics.** In Table 1, we compare the distribution of domains in OLC to that of the Pile (Gao et al., 2020), a popular pretraining corpus that includes data under copyright restrictions (e.g., Books, web crawl).[4] These statistics convey a number of research challenges when working with OLC. First, despite our best efforts to collect as much public domain or permissively licensed data as possible, the size of OLC is still 31% smaller than the Pile. In addition, while the majority of the Pile is sourced

---

[3]https://www.kaggle.com/datasets/stackoverflow/stackoverflow
[4]This comparison also dovetails with our comparison to Pythia, a model trained on the Pile (§4).

from scientific text, web crawl, and books, OLC is dominated by code, scientific text, and legal text. This highlights that models designed for use outside these specific domains may struggle and require special techniques for extreme domain generalization. More analysis can be found in §B.

## 3.2 ADDING A NONPARAMETRIC COMPONENT

To incorporate the nonparametric component, we experiment with two widely-used retrieval methods: the $k$-nearest neighbors LM ($k$NN-LM; Khandelwal et al., 2020) and the retrieval-in-context approach (RIC-LM; Shi et al., 2023; Ram et al., 2023). Each approach constructs a datastore from the raw text data offline, and then uses it on-the-fly at inference time.

The **$k$-nearest neighbors language model ($k$NN-LM)** interpolates the next-token probability distribution from a parametric LM with a nonparametric distribution based on every token that is stored in a datastore. Given a text dataset consisting of $N$ tokens $c_1...c_N$, a datastore is built by creating a key-value pair for every token $c_i$ ($1 \leq i \leq N$); a value is $c_i$ and a key $k_i$ is $...c_{i-1}$, a prefix preceding $c_i$. At test time, given an input prefix $x$, the nonparametric distribution is computed by:

$$P_{k\text{NN}}(y \mid x) \propto \sum_{(k,v)\in\mathcal{D}} \mathbb{I}[v = y]\exp\left(-d(\text{Enc}(k), \text{Enc}(x))\right).$$

Here, $\text{Enc}$ is an encoder that maps a text into $\mathbb{R}^h$ and $d : \mathbb{R}^h \times \mathbb{R}^h \to \mathbb{R}$ is a distance function, where $h$ is the hidden dimension. We follow Khandelwal et al. (2020) and use the output vector from the last layer of the transformers in the parametric LM as $\text{Enc}$, and L2 distance as $d$. The final model takes the $k$NN-LM output and interpolates it with the output from the parametric LM: $\lambda P_{\text{LM}}(y \mid x) + (1 - \lambda)P_{k\text{NN}}(y \mid x)$, where $\lambda$ is a fixed hyperparameter between 0 and 1.

As an alternative to $k$NN-LM, the **retrieval-in-context language model (RIC-LM)** retrieves text blocks from a datastore and feeds them to the parametric LM in context. Given a dataset consisting of $N$ tokens $c_1...c_N$, an index $\mathcal{D}$ is constructed by splitting the data into text blocks $b_1...b_M$, optionally with a sliding window. At test time, given an input prefix $x$, RIC-LM retrieves the most similar paragraph to the prefix $\hat{p} = \arg\max_{b\in\mathcal{D}} \text{sim}(b, x)$ and concatenates it before the prefix to produce $P_{\text{LM}}(y \mid \hat{b}, x)$. Here, $\text{sim}$ is a function that computes a similarity score between two pieces of text; we use BM25 following Ram et al. (2023) who show that BM25 outperforms alternative dense retrieval.

**Comparison between $k$NN-LM and RIC-LM.** $k$NN-LM and RIC-LM differ in how the nonparametric component influences the output. In $k$NN-LM, it directly impacts the output distribution, while in RIC-LM, it indirectly influences the output by affecting the input to the parametric model. This indicates $k$NN-LM would intuitively benefit more from a datastore; however, RIC-LM interacts more easily with a parametric model (i.e., it is applicable to a black-box LM since probabilities do not need to be accessed) and offers better speed and memory efficiency (explored in Appendix D.2). Empirical comparisons between kNN-LM and RIC-LM have been largely unexplored. Our experiments (§4.2) present a series of such comparisons, with varying sizes of the datastore, and with and without distribution shift.

**Datastore.** We assume in-distribution data for each test domain is available at inference time, and construct an in-domain datastore for each test domain based on its training data; future work may investigate building a single datastore that includes all domains. For datasets from the PILE, we consider 10% of the training data. For $k$NN-LM, each datastore is capped to 1 billion tokens due to the resource constraints. More implementation details, statistics and hyperparameter values for the datastores are reported in §C.

**Attribution and opt-out.** Since elements in the datastore that contribute to the model prediction are transparent, both $k$NN-LM and RIC-LM offer inherent attributions. Moreover, data removed from the datastore is guaranteed not to contribute to any model predictions, allowing data owners to remove their data at the level of individual examples. Both are unique characteristics of nonparametric language models. While prior work studies post-hoc attribution to the data used for training model parameters (Koh & Liang, 2017; Han et al., 2023) and removing the effect of specific training examples from parameteric models (Cao & Yang, 2015; Jang et al., 2023b), they lack guarantees and are difficult to scale.

Implementation details, statistics and hyperparameter values for the datastores are reported in §C.

Table 2: Perplexity (the lower the better) of the parametric-only SILO trained on $\overline{\text{PD}}$, $\overline{\text{PDSW}}$, and $\overline{\text{PDSWBY}}$ (without a datastore), compared to Pythia-1.4B, a model trained with similar amounts of compute but on mostly non-permissive data. We use ▆, ▆, and ▆ to indicate text that is in-domain, out-of-domain, or out-of-domain but has relevant data in-domain (e.g., high-risk Github code vs. our permissive Github code). Reported on the test data; see Table 11 (§D) for results on the validation data. **Our parametric LMs are competitive to Pythia in-domain but fall short out-of-domain.**

| Eval data | $\overline{\text{PD}}$ | $\overline{\text{PDSW}}$ | $\overline{\text{PDSWBY}}$ | Pythia |
|---|---|---|---|---|
| FreeLaw | 5.3 | 5.7 | 6.5 | 5.6 |
| Gutenberg | 15.2 | 12.5 | 14.1 | 13.1 |
| HackerNews | 38.0 | 13.7 | 14.5 | 13.3 |
| Github | 13.5 | 2.7 | 2.8 | 2.4 |
| NIH ExPorter | 28.2 | 19.2 | 15.0 | 11.1 |
| PhilPapers | 31.7 | 17.6 | 15.0 | 12.7 |
| Wikipedia | 28.9 | 20.3 | 11.3 | 9.1 |
| CC News | 34.0 | 23.3 | 21.2 | 12.0 |
| BookCorpus2 | 25.3 | 19.2 | 19.6 | 13.2 |
| Books3 | 27.2 | 19.3 | 18.6 | 12.6 |
| OpenWebText2 | 37.8 | 21.1 | 18.8 | 11.5 |
| Enron Emails | 18.6 | 13.2 | 13.5 | 6.9 |
| Amazon | 81.1 | 34.8 | 37.0 | 22.9 |
| MIMIC-III | 22.3 | 19.0 | 15.5 | 13.1 |
| Average | 29.1 | 17.3 | 16.0 | 11.4 |

## 4 EXPERIMENTAL RESULTS

We benchmark our models using *language modeling perplexity* on 14 domains as shown in Table 2 (details provided in §C.3). Experiments consider both in-domain and out-of-domain data and different levels of OLC at different levels of legal risk, e.g., CC-News, BookCorpus2, Books3 and Amazon reviews are mostly copyrighted, Github is mostly not permissively licensed,[5] and Enron Emails and MIMIC-III include private text. We also evaluate on *zero-shot downstream performance* on ten text classification datasets in §4.2 (details provided in §C.4).

We first evaluate the parametric-only component of SILO trained on the OLC (§4.1), and then show the effect of adding a datastore that may contain high-risk text (§4.2). For all experiments, we use the 1.4B Pythia model (Biderman et al., 2023) as a baseline because it is trained with a similar amount of compute (data size and model parameters), but is trained on mostly high-risk data.[6]

### 4.1 RESULTS: PARAMETRIC COMPONENT

**Main results.** Table 2 reports performance of our 1.3B base LMs trained on varying levels of permissively-licensed data—$\overline{\text{PD}}$, $\overline{\text{PDSW}}$, and $\overline{\text{PDSWBY}}$—as well as Pythia. Overall, our LMs are competitive with Pythia despite using permissive data only. They are roughly equal quality on in-domain data, e.g., FreeLaw and Gutenberg, HackerNews in the case of $\overline{\text{PDSW}}$ and $\overline{\text{PDSWBY}}$, and Wikipedia in the case of $\overline{\text{PDSWBY}}$. The largest gaps occur on data that is in-domain for Pythia but out-of-domain for our model, e.g., news, books, OpenWebText, and emails, and Wikipedia in the case of models besides $\overline{\text{PDSWBY}}$. This illustrates the extreme domain generalization challenge that is present when training on only permissive data, as we hint in §3.1.

**Gaps from Pythia align with a degree of domain shift.** The similarity of an evaluation domain to a domain of the OLC strongly correlates with the performance gaps between SILO and Pythia. To show this, we compute the Pearson correlation between 1) the maximum $n$-gram overlap between an OLC domain and the Pile validation domains (from §B.2 and 2) the perplexity difference between the Pythia model and our $\overline{\text{PDSW}}$ model, normalized by the performance of the $\overline{\text{PDSW}}$ model. We find a strong negative correlation between these metrics ($r$=-0.72, $p < 0.005$), indeed indicating that the

---

[5]Kocetkov et al. (2023) estimates about 13% of the Github data is under MIT, Apache, and BSD.
[6]We use the model checkpoint from https://huggingface.co/EleutherAI/pythia-1.4b-deduped-v0.

Table 3: Perplexity (the lower the better) of parametric LMs (Prm-only), $k$NN-LM, and RIC-LM. % in parentheses indicate a reduction in the gap between the parametric-only SILO and Pythia. As in Table 2, ■ indicates in-domain; ■ indicates out-of-domain; ■ indicates out-of-domain but has relevant data in-domain, all with respect to the training data of the parametric LM. Reported on the test data; see Table 12 for results on the validation data. See Table 9 for the statistics of the datastore. **Adding a datastore, with $k$NN-LM, effectively reduces the gap between SILO and Pythia.**

| Eval data | SILO (PDSW) | | | Pythia |
|---|---|---|---|---|
| | Prm-only | $k$NN-LM | RIC-LM | Prm-only |
| Github | 2.7 | 2.4 (-100%) | 2.4 (-100%) | 2.4 |
| NIH ExPorter | 19.2 | 15.0 (-52%) | 18.5 (-9%) | 11.1 |
| Wikipedia | 20.3 | 14.5 (-52%) | 19.4 (-8%) | 9.1 |
| CC News | 23.3 | 8.0 (-135%) | 16.8 (-58%) | 12.0 |
| Books3 | 19.3 | 17.4 (-28%) | 18.6 (-10%) | 12.6 |
| Enron Emails | 13.2 | 5.9 (-116%) | 9.9 (-68%) | 6.9 |
| Amazon | 34.9 | 26.0 (-75%) | 33.7 (-10%) | 23.0 |
| MIMIC-III | 19.0 | 6.6 (-210%) | 15.6 (-58%) | 13.1 |
| Average | 19.0 | 12.0 (-91%) | 16.9 (-27%) | 11.3 |

more dissimilar an evaluation domain is from the OLC domains, the better Pythia does relative to SILO; see §D (Figure 6) for a scatter plot.

More ablations, including the effect of upsampling low-resource data and the effect of explicit source code, are provided in §D.1.

## 4.2 RESULTS: ADDING THE NONPARAMETRIC COMPONENT

Since building legally permissive LMs poses a challenge of extreme domain generalization, our next question is whether using an in-domain, nonparametric datastore can reduce the gap. This section reports results with the LM trained on the PDSW subset of OLC; see §D.2 for results of models trained on PD or PDSWBY. All models are evaluated on a subset of 8 out-of-domain datasets: Github, NIH ExPorter, Wikipedia, CC News, Books3, Enron Emails, Amazon, and MIMIC-III.

**Main results.** Table 3 shows adding the datastore with either $k$NN-LM- or RIC-LM-based retrieval improves performance over just using the parameteric component on all domains, but $k$NN-LM is more effective than RIC-LM. In most domains, $k$NN-LM reduces the gap between SILO and Pythia by more than 50% (on NIH ExPorter, Wikipedia, Amazon) or even outperforms Pythia (on Github, CC News, Enron Emails, MIMIC-III). Books3 is the domain with the least benefit, although $k$NN-LM still reduces the gap by 28%.

**Impact of scaling the datastore.** Figure 2 demonstrates that both $k$NN-LM and RIC-LM-based retrieval consistently improve performance as the datastore size increases, with a strong log-linear trend. However, $k$NN-LM improves performance more rapidly than RIC-LM does, consistently over all datasets. Extrapolating the trend suggests that, on the domains that SILO has not outperformed Pythia yet, scaling the datastore even further (with $k$NN-LM retrieval) may enable it to match Pythia.

**Why does $k$NN-LM outperform RIC-LM?** Our next question is why $k$NN-LM is better than RIC-LM—is it (a) because $k$NN-LM is better than RIC-LM in general, or (b) because $k$NN-LM generalizes out-of-domain better than RIC-LM does? Our further analysis in §D.2 (Figure 5) reveals that it is both. $k$NN-LM overall outperforms and scales better than RIC-LM even when test domains are in-domain, supporting (a). The gains are larger when test domains are out-of-domain than when test domains are in-domain, supporting (b). We also find that the robustness of $k$NN-LM can be explained by the encoder for a nonparametric distrbution $P_{\text{kNN}}(y \mid x)$ being more robust to a distribution shift than the model that outputs $P_{\text{LM}}(y \mid x)$; see §D.2 (Figure 4) for the details.

Altogether, our analysis highlights two promising directions to further reduce the gap: (1) scaling the datastore beyond 1 billion tokens, e.g., at the scale of trillions of tokens as in Borgeaud et al. (2022), and (2) improving the robustness of the model by improving nonparametric techniques (Zhong et al., 2022; Lan et al., 2023; Min et al., 2023; Huang et al., 2023b; Izacard et al., 2022).

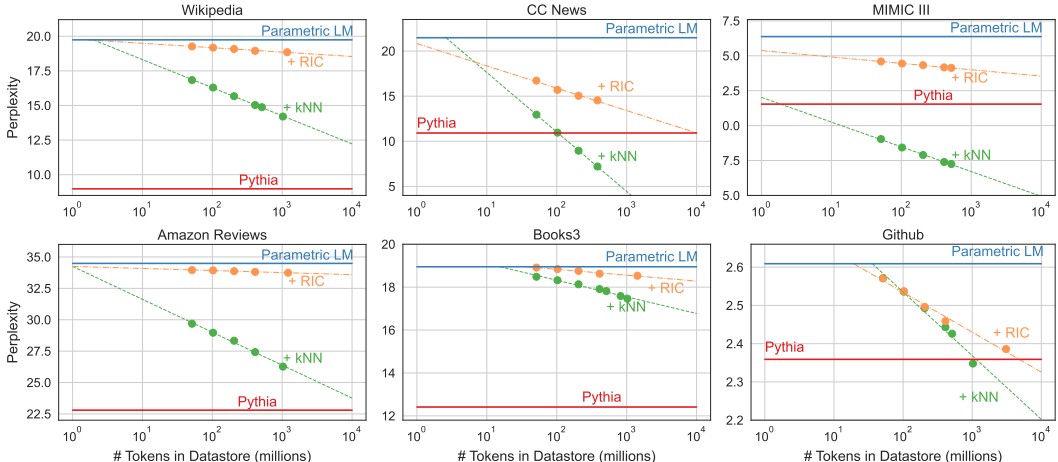

Figure 2: Impact of scaling the datastore of SILO ([PDSW]). Perplexity on random 128K tokens from the validation data reported. The rightmost dots for $k$NN-LM and RIC-LM in each figure correspond to the final models used in Table 3. **Scaling the test-time datastore consistently improves performance over all domains.**

**Downstream task performance.** In order to verify if our findings on language modeling perplexity transfer to downstream tasks, we evaluate zero-shot performance of SILO and Pythia on ten text classification datasets whose domains are not covered by OLC. All models use PMI (Holtzman et al., 2021) for a better calibration of model outputs, and we use $k$NN-Prompt (Shi et al., 2022) for applying $k$NN-LM for downstream tasks. See §C.4 for the details. Table 4 demonstrates that our earlier findings hold on all ten datasets: the parametric-only SILO largely underperforms Pythia; however, adding a datastore greatly improves performance, allowing performance of SILO to match that of Pythia.

**Comparison in runtime speed.** Table 16 (§D.2) provides a comparison of the runtime speed of the parametric LM, RIC-LM, and $k$NN-LM. There is a clear tradeoff between performance and speed: both RIC-LM and $k$NN-LM are considerably slower than the parametric LM, and a larger datastore and more accurate nearest-neighbor search leads to better performance and slower inference. We leave improving the runtime efficiency of nonparametric approaches for future work.

Table 4: **Zero-shot performance on ten classification datasets.** $k$NN-LM allows performance of SILO to match that of Pythia by using a datastore. ▨ indicates in-domain; ▨ indicates out-of-domain.

| Eval | SILO ([PDSW]) | | Pythia |
| | Prm-only | $k$NN-LM | Prm-only |
| --- | --- | --- | --- |
| AGN | 63.3 | 79.5 | 71.2 |
| Dbpedia | 36.9 | 41.3 | 39.1 |
| SST-2 | 57.1 | 74.4 | 79.7 |
| MR | 55.1 | 79.0 | 79.9 |
| RT | 55.3 | 67.9 | 80.4 |
| CR | 64.6 | 83.1 | 80.3 |
| Yelp | 62.8 | 84.5 | 84.7 |
| Amz | 60.4 | 82.9 | 80.3 |
| RTE | 56.0 | 55.2 | 53.7 |
| HYP | 58.5 | 63.1 | 58.5 |
| Avg | 57.0 | 71.1 | 70.8 |

## 4.3 EXAMPLES OF DATA ATTRIBUTION AND OPT-OUT

As discussed in §2, the design of SILO can better align with various data-use regulations by providing mechanisms for data attribution during inference and for data owners to remove their data from the model at any time.

**Data opt-out.** To showcase the impact of opt-out on model performance, we conduct experiments with J. K. Rowling's Harry Potter series. We first identify all seven Harry Potter books from the Books3 corpus of the Pile. For each book, we calculate the perplexity of SILO using two 1B-token datastores on Books3, but one including the remaining six Harry Potter books and the other excluding any Harry Potter books. This experiment aims to see whether excluding Harry Potter books from the former datastore can reduce the likelihood of generating the left-out book.

Table 6: **Attribution examples on Harry Potter books.** We show the top-1 retrieved context of SILO (PDSW). **Red underlined text** indicates the next token that immediately follows the prefix. In both examples, the test data is from the sixth novel and the retrieved context is from the fourth novel in the Harry Potter series. In the series, *Azkaban* is the notorious wizarding prison, and the *green light* is a distinct characteristic of the Killing Curse, *Avada Kedavra*.

---

**Test Prefix** 'I - what - *dragons*?' spluttered the Prime Minister. 'Yes, three,' said Fudge. 'And a sphinx. Well, good day to you.' The Prime Minister hoped beyond hope that dragons and sphinxes would be the worst of it, but no. Less than two years later, Fudge had erupted out of the fire yet again, this time with the news that there had been a mass breakout from **Test Continuation** **Azkaban**.
**Retrieved Prefix** 'D' you know Crouch, then?' said Harry. Sirius' face darkened. He suddenly looked as menacing as the night when Harry had first met him, the night when Harry had still believed Sirius to be a murderer. 'Oh, I know Crouch all right,' he said quietly. 'He was the one who gave me the order to be sent to **Retrieved Continuation** **Azkaban** - without a trial.'

**Test Prefix** Terror tore at Harry's heart... he had to get to Dumbledore and he had to catch Snape... somehow the two things were linked... he could reverse what had happened if he had them both together... Dumbledore could not have died... (...) Harry felt Greyback collapse against him; with a stupendous effort he pushed the werewolf off and onto the floor as a jet of **Test Continuation** **green** light
**Retrieved Prefix** Voldemort was ready. As Harry shouted, "Expelliarmus!" Voldemort cried, "Avada Kedavra!" A jet of **Retrieved Continuation** **green** light issued from Voldemort's wand

---

Table 5 shows the results. SILO with Harry Potter books in the datastore (third column) effectively improves perplexity for all seven books, closing the gap between the PDSW model (first column) and Pythia (fourth column). However, when the Harry Potter books are removed from the datastore (second column), the perplexity gets worse, approaching that of the parametric-only LM. This illustrates that eliminating the effect of the Harry Potter books from the model substantially reduces the likelihood of generating the leave-out book.

**Attribution.** To show the attribution feature of our model, Table 6 provides qualitative examples on the top-1 context retrieved by SILO. The model is able to assign a high probability to the ground truth token by retrieving highly relevant context. It achieves this by leveraging the unique characteristics of the text within the datastore, such as recognizing that *Azkaban* refers to the prison and *green light* is associated with the *Killing Curse* in the Harry Potter books.

Table 5: **The effect of data opt-out**, reporting perplexity. *w/ HP* and *w/o HP* indicate that the datastore includes or excludes Harry Potter books, respectively. The number (1 to 7) indicates a different book from the Harry Potter series used as the eval data; this eval book is not included in the datastore in any case. ■ indicates in-domain; ■ indicates out-of-domain.

| Eval | SILO (PDSW) | | | Pythia |
| | Prm-only | $k$NN-LM w/o HP | $k$NN-LM w/ HP | Prm-only |
|---|---|---|---|---|
| 1 | 15.9 | 15.2 | 13.0 | 9.6 |
| 2 | 17.7 | 16.7 | 12.4 | 10.0 |
| 3 | 16.5 | 15.6 | 11.4 | 9.5 |
| 4 | 17.7 | 16.8 | 12.9 | 10.1 |
| 5 | 17.8 | 16.9 | 13.2 | 10.2 |
| 6 | 17.4 | 16.5 | 12.8 | 10.1 |
| 7 | 18.8 | 17.8 | 15.1 | 10.9 |
| Avg | 17.4 | 16.5 | 12.9 | 10.1 |

More qualitative examples on Github, news and emails are illustrated in Table 17 in §D.2. They highlight that a nonparametric approach addresses specific legal risks that we have discussed earlier, e.g., it offers per-token attribution for free, and can provide a copyright notice when part of copyrighted text is being used for the probability distribution.

## 5 CONCLUSION

We introduce SILO, a language model that mitigates legal risk by learning parameters only on lower-risk, permissively licensed data (OPEN LICENSE CORPUS), and using an unrestricted nonparametric datastore during inference. Our approach allows the model designer to use higher-risk data without training on it, supports per-token data attribution, and enables data produces to opt-out from the model by removing content from the datastore. Experiments on language modeling perplexity and downstream text classification show that using only the parametric component of SILO is competitive on domains covered by OPEN LICENSE CORPUS, but falls short out-of-domain, highlighting the challenge of extreme domain generalization. We then show that adding a nonparametric datastore (with $k$NN-LM retrieval) successfully addresses this challenge, significantly reducing the gap (or even outperforming) the Pythia baseline that is trained unrestrictedly. Further ablations and analyses point to a number of exciting avenues for future work, which we highlighted in §E. Altogether, our results suggest that providing technical solutions to mitigate legal risk in AI systems is a promising and important direction for future research.

ACKNOWLEDGMENTS

We thank Peter Henderson and Mark Lemley for discussing the legality of LMs, and Kyle Lo for feedback on our dataset and license taxonomy. We thank Mitchell Wortsman for help with setting up compute and model training. We thank Chris Callison-Burch, James Grimmelmann, Tatsunori Hashimoto, Peter Henderson, Nikhil Kandpal, Pang Wei Koh, Mark Lemley, Kyle Lo, Fatemeh Mireshghallah, Sewoong Oh and Rulin Shao for valuable feedback on the project and the paper. We thank Matthijs Douze, Gergely Szilvasy, and Maria Lomeli for answering questions about FAISS, and Dirk Groeneveld for giving early access to the deduplication script. Sewon Min is supported by the J.P. Morgan Ph.D. Fellowship. Suchin Gururangan is supported by the Bloomberg Data Science Ph.D. Fellowship. Eric Wallace is supported by the Apple Scholars in AI/ML Fellowship. We thank Stability AI for providing compute to train the LMs in this work. At UW, this work supported in part by the Office of Naval Research under MURI grant N00014-18-1-2670, the DARPA MCS program through NIWC Pacific (N66001-19-2-4031), NSF IIS-20444660, and gifts from AI2.

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

## A   ADDITIONAL BACKGROUND

**Legality of language models.**   The legality of training LMs this way has become a subject of intense debate, with numerous lawsuits being filed in the United States, United Kingdom, and beyond (Gershgorn, 2021; Metz, 2022; Vincent, 2023; De Vynck, 2023; Silverman et al. v. Meta Platforms, Inc., 2023; J.L. et al. v. Alphabet Inc., 2023; Silverman et al. v. OpenAI, Inc., 2023; Tremblay et al. v. OpenAI, 2023). While the outcome of the lawsuits is uncertain, it is likely that such legal issues will continue to be a major factor in future LMs, especially since each country has its own data regulations. For example:

- In the United States, the *fair use doctrine* allows the public to use copyrighted data in certain cases, even without a license (Henderson et al., 2023). Deciding whether or not a model's use of copyrighted work constitutes fair use involves multiple dimensions, including whether the trained model is intended for commercial use, whether or not the work is factual or creative, the amount of the copyright content used, and the value of the copyrighted work. There are claims that using parametric language models for *generative* use-cases does *not* constitute fair use, because the technology may output the copyrighted text verbatim (Lemley & Casey, 2020), which also has been shown empirically (Carlini et al., 2021; 2023; Kandpal et al., 2022; Chang et al., 2023). This is in contrast to *transformative* technologies, such as classifiers, which may use the copyrighted text but do not directly generate content, which the fair use doctrine favors. We refer readers to Henderson et al. (2023) for a more comprehensive discussion.

- The *General Data Protection Regulation (GDPR)* is a comprehensive data protection and privacy law in the European Union (EU). It grants individuals more control over their data by regulating organizations and businesses. The obligations include (1) obtaining consent from users before processing their data, (2) providing transparency about data processing, (3) ensuring data security, and (4) allowing individuals to access, correct, and erase their data. GDPR has global impact, as many international companies handle EU citizens' data. While it is under debate how GDPR is applied to training language models, compliance with GDPR is expensive (e.g., requiring retraining for every data correction or erasure). See Zhang et al. (2023) for more discussion on challenges for compliance with the GDPR's *Right to Erasure* (and the *Right to be Forgotten* in general).

The goal of our work is not to weigh in on legal discussions; instead, we study the feasibility of developing technologies that explicitly manage legal risk. In particular, our technique places all copyrighted data in a nonparametric datastore. While the data is still used in service of a generative model, restricting copyrighted data in a datastore and providing instance-level attribution and data opt-out can increase the likelihood of a successful fair use defense (Henderson et al., 2022).[7] Moreover, GDPR's requirement regarding user data access, correction, and erasure aligns well with the capabilities of the datastore. Attribution and opt-out are fundamental features of our model (§3.2). This is in contrast to other techniques like post-hoc training data attribution (Koh & Liang, 2017; Han et al., 2023) and the removal of the effect of particular training examples from parameters (Cao & Yang, 2015; Jang et al., 2023b), which lack inherent guarantees and are hard to scale.

## B   DETAILS OF OPEN LICENSE CORPUS

In this section, we describe details of OPEN LICENSE CORPUS (§3.1): our taxonomy of data licenses (§B.1) and data analysis (§B.2).

**A disclaimer.**   The license taxonomy and categorization of texts that we present is by no means perfect, and OLC should not be considered a universally safe-to-use dataset. The license associated with a document may be time- and country-dependent, e.g., Gutenberg books (Project Gutenberg) are public domain in the United States, but some of them may still have copyright attached outside of the United States. Moreover, other legal constraints (e.g., the Digital Millenium Copyright Act)[8] may prohibit the use of a data source despite a permissive data license. Finally, we do not explicitly filter out personally identifiable information from the corpus, so it is possible that certain subsets still

---

[7]Our model on its own does not entirely remove legal risk. Rather, it provides functionalities that, when used appropriately, lower legal risk and strengthen a fair use defense. See §E for a discussion.

[8]https://www.copyright.gov/dmca/

Table 7: **Overview statistics of OLC**. $\overline{\text{PD}}$, $\overline{\text{SW}}$, and $\overline{\text{BY}}$ indicates public domain data, data under permissive software licenses, and data under attribution licenses, respectively. Some corpora contain a mixture of different licenses (e.g., Pile of Law and S2ORC), which we split into subsets based on per-document licenses. BPE tokens are based on the GPT-NeoX tokenizer (Black et al., 2022).

| Domain | Sources | Specific License | # BPE Tokens (B) |
|---|---|---|---|
| Legal | $\overline{\text{PD}}$ Case Law, Pile of Law (PD subset) | Public Domain | 27.1 |
| | $\overline{\text{BY}}$ Pile of Law (CC BY-SA subset) | CC BY-SA | 0.07 |
| Code | $\overline{\text{SW}}$ Github (permissive) | MIT/BSD/Apache | 58.9 |
| Conversational | $\overline{\text{SW}}$ HackerNews, Ubuntu IRC | MIT/Apache | 5.9 |
| | $\overline{\text{BY}}$ Stack Overflow, Stack Exchange | CC BY-SA | 21.3 |
| Math | $\overline{\text{SW}}$ Deepmind Math, AMPS | Apache | 3.5 |
| Science | $\overline{\text{PD}}$ ArXiv abstracts, S2ORC (PD subset) | Public Domain | 1.2 |
| | $\overline{\text{BY}}$ S2ORC (CC BY-SA subset) | CC BY-SA | 70.3 |
| Books | $\overline{\text{PD}}$ Gutenberg | Public Domain | 2.9 |
| News | $\overline{\text{PD}}$ Public domain news | Public Domain | 0.2 |
| | $\overline{\text{BY}}$ Wikinews | CC BY-SA | 0.01 |
| Encyclopedic | $\overline{\text{BY}}$ Wikipedia | CC BY-SA | 37.0 |

pose privacy risks despite being permissively licensed. We encourage users of OLC to consult a legal professional on the suitability of each data source for their application.

## B.1 TAXONOMY OF DATA LICENSES

Determining what data one is permitted to use from a copyright perspective is an ongoing topic of debate, and is context- and country-dependent (Henderson et al., 2023). In this paper, we take a conservative approach where we train models using only text with the most permissible licenses, thus enabling widespread downstream use. Concretely, we focus on four broad categories:

- **Public domain** ($\overline{\text{PD}}$) text has no restrictions. This includes texts whose intellectual property rights have expired (e.g., the works of William Shakespeare) or been expressly waived by the creator (e.g., CC0-licensed scientific papers).

- **Permissively licensed software** ($\overline{\text{SW}}$) including MIT, Apache, and BSD software are quite permissive to use. Unlike public domain text, these licenses typically carry some basic stipulations such as requiring one to include a copy of the original license (although, it is debatable whether it is still required when the associated text is used as data or treated as a software). The code is otherwise free to use, and code is generally well protected by fair use clauses (Lemley & Casey, 2020).

- **Attribution licenses** ($\overline{\text{BY}}$) such as Creative Commons Attribution (CC-BY) are free to use as long as "credit is given to the creator." For example, if a journalist quotes an article from Wikipedia (a CC-BY source), then they must provide a form of citation, link, or attribution back to the original source. In the context of machine learning, it is not clear what an attribution would constitute. For example, under one interpretation, every LM generation should include a complete list of sources that contributed highly to it (Henderson et al., 2023). In this paper, we take a conservative approach and do not include $\overline{\text{BY}}$ data in the main experiments, but still include the $\overline{\text{BY}}$ data for future use as well as for ablations, since $\overline{\text{BY}}$ data is generally considered quite permissive.

- **All other data** that is not in one of the above three categories is assumed to be non-permissive. This includes: any text that is explicitly protected by copyright or licenses that are non-commercial (e.g., CC-NC), any software without clear MIT, BSD, or Apache licenses, and any generic web-crawled data where the license or copyright information may be unclear.

In §3.1, we train the models on varying subsets of licenses—from $\overline{\text{PD}}$ and $\overline{\text{PDSW}}$ to $\overline{\text{PDBYSW}}$—to accommodate different risk tolerances.

## B.2 DATA ANALYSIS

To study the composition of OLC, we perform an $n$-gram based analysis of OLC domains against the validation data of the Pile. This enables us to better understand the dataset's domain shifts. We sample up to 10M tokens in each data source of OPEN LICENSE CORPUS and each subset of the Pile validation data. We then compute unigrams and bigrams across domains in both datasets (ignoring stopwords), and only consider $n$-grams that appear at least three times in each domain. For each validation domain of the Pile, we examine the *maximum* $n$-gram overlap across all OLC domains.

OLC domains have substantially less overlap with the validation data as compared to the Pile training domains: on average, the overlap between OLC domains and the validation domains is just 17%±9%, versus 28%±14% for the Pile training data. However, we find a large variance in overlap statistics across domains in OLC; we display the full matrix of $n$-gram overlap in Table 18. These results provide further evidence that models trained on OLC must handle larger domain shifts at test time than models trained on the Pile.

## C EXPERIMENTAL DETAILS

### C.1 DETAILS ON THE PARAMETRIC COMPONENT

**LM architecture.**   We use 1.3B-parameter transformer LMs based on the LLaMA architecture (Touvron et al., 2023) as implemented in OpenLM.[9] Table 8 reports the hyperparameters for the parametric component of SILO. We keep these hyperparameters fixed for all parametric models that we report in this paper. We use the GPT-NeoX-20B tokenizer (Black et al., 2022), with 50432 BPE types.

Table 8: Basic hyperparameters for the parametric component of SILO.

| Model | #L | #H | $d_{model}$ | LR | Batch |
|-------|----|----|-------------|-----|-------|
| 1.3B | 24 | 16 | 2048 | 1e-3 | 2.6M |

**Training.**   Each model is trained with 128 A100 GPUs across 16 nodes. We use 2,048 token sequences that are packed across document boundaries, and we pre-pend a beginning-of-text token to every document. We use weight decay of 0.1, the Adam optimizer with $\beta_2 = 0.95$, 2,000 steps of warmup, with a cosine learning rate scheduler. Following Muennighoff et al. (2023), we train for multiple epochs in each dataset, tracking validation perplexity every 10B tokens, and perform early stopping. As a result, we train our PD, PDSW and PDSWBY models for 60B, 250B, and 350B tokens in total, respectively.

**Domain re-weighting.**   Since the distribution of OLC is highly skewed (§3.1), we perform a simple upweighting scheme where we upsample all data that accounts for less than 5% by a factor of $3\times$, which we found to work well after a sweep of different settings. More sophisticated domain weighting strategies (Xie et al., 2023) are of interest but beyond the scope of this work.

### C.2 DETAILS OF ADDING THE NONPARAMETRIC COMPONENT

For $k$NN-LM, each datastore consists of up to 1 billion $h$-dimensional vectors ($h =$2,048). We build an index for fast nearest neighbor search using FAISS (Johnson et al., 2019). We use IndexIVFPQ which quantizes vectors into 64-bytes and clusters them into 4,096 centroids, learned from 1 million sampled vectors, following Khandelwal et al. (2020). Instead of recomputing the exact L2 distance using the original embeddings, we use the L2 distance beteen quantized vectors returned by the FAISS index (ablations in Appendix D.2). Since their scale is not preserved, we use $\frac{d(\mathbf{x}_q, \mathbf{y}_q)}{\tau}$ as a proxy of $d(\mathbf{x}, \mathbf{y})$, where $\mathbf{x}_q$ and $\mathbf{y}_q$ are vectors quantized from $\mathbf{x}$ and $\mathbf{y}$. Hyperparameters, including $k$, $\lambda$, and $\tau$, are chosen based on the validation data in a domain-specific manner.

For **RIC-LM**, each datastore consists of text blocks with a length of 1,024 and a sliding window of 512. We use BM25 from Pyserini (Lin et al., 2021). Appendix D.2 report ablations on different

---

[9]https://github.com/mlfoundations/open_lm

Table 9: Datastore statistics as well as hyperparameter values for $k$NN-LM. Underline indicates exact nearest neighbor search (instead of approximate) was performed for $k$NN-LM because the datastore is small enough. Hyperparameters are chosen based on the validation data of each domain.

| Data | RIC-LM | | $k$NN-LM | | | |
|------|----------|----------|----------|-----|--------|------|
| | # tokens | # blocks | # tokens | $\lambda$ | $k$ | $\tau$ |
| Github | 3084.3M | 6.0M | 1024.0M | 0.2 | 128 | 10.0 |
| NIH ExPorter | 72.2M | 0.1M | 72.2M | 0.3 | 32,768 | 20.0 |
| Wikipedia | 1177.9M | 2.3M | 1024.0M | 0.3 | 4,096 | 20.0 |
| CC News | 382.2M | 0.7M | 382.2M | 0.7 | 4,096 | 20.0 |
| Books3 | 1424.7M | 2.8M | 1024.0M | 0.2 | 4,096 | 25.0 |
| Enron Emails | 45.0M | 0.1M | 45.0M | 0.5 | 4,096 | 1.0 |
| Amazon | 1214.3M | 2.4M | 1024.0M | 0.5 | 32,768 | 20.0 |
| MIMIC-III | 519.5M | 1.0M | 519.5M | 0.7 | 1,024 | 15.0 |

Table 10: Datastore and hyperparameter values for $k$NN-LM evaluated on downstream tasks. Hyperparameters (the choice of datastore, $\lambda$, $k$ and $\tau$) are chosen based on the validation data.

| Task | Datastore | $\lambda$ | $k$ | $\tau$ | Template | Label Words |
|------|-----------|-----------|-----|--------|----------|-------------|
| AGN | Wikipedia | 0.8 | 1024 | 3 | The topic of the text is | politics, sports, business, technology |
| Dbpedia | Amazon | 0.9 | 4096 | 3 | The topic of the text is | company, school, artist, athlete ... (14 in total) |
| SST-2 | IMDB | 0.9 | 8192 | 3 | It was | great, terrible |
| MR | IMDB | 0.9 | 8192 | 3 | It was | great, terrible |
| RT | IMDB | 0.9 | 8192 | 3 | It was | great, terrible |
| CR | Amazon | 0.7 | 1024 | 1 | It was | great, terrible |
| Yelp | IMDB | 0.9 | 8192 | 3 | It was | great, terrible |
| Amz | IMDB | 0.9 | 8192 | 3 | It was | great, terrible |
| RTE | Amazon | 0.9 | 1024 | 3 | true or false? answer: | true, false |
| HYP | CC News | 0.1 | 1024 | 1 | neutral or partisan? answer:? | neutral, partisan |

implementations of RIC-LM besides the method in §3.2, including different ways of choosing the retrieved context and using multiple text blocks.

Table 9 reports the datastore statistics for both RIC-LM and $k$NN-LM, as well as hyperparameter values for $k$NN-LM ($\lambda, k, \tau$). Due to the resource constraints, the datastore size is capped to up to 10% of the PILE training data (and to 1024.0M tokens in the case of $k$NN-LM), but future work can investigate further scaling the datastore.

## C.3 DETAILS OF LANGUAGE MODELING PERPLEXITY EVALUATION

We benchmark our models using language modeling perplexity on 14 domains that represent both in-domain and out-of-domain data with respect to different levels of OLC. This includes: public-domain legal documents from the **FreeLaw** Project subset of the the Pile (Gao et al., 2020), a held-out collection of books from the **Gutenberg** collection (Project Gutenberg), conversational text from the **Hacker News** subset of the Pile, held-out code files from the **Github** subset of the Pile (most of which are non-permissive licensed), scientific text of NIH Grant abstracts that are taken from the **NIH ExPorter** subset of the PILE, philosophy papers taken from the **PhilPapers** of the PILE, held-out English **Wikipedia** articles from the PILE, news articles from **CC-News** (Mackenzie et al., 2020), books from **BookCorpus2** which is an expanded version of Zhu et al. (2015), books from **Books3** by Presser (2020), random web-crawled pages from **OpenWebText2** (Gokaslan & Cohen, 2019; Gao et al., 2020), emails from the **Enron Emails** corpus (Klimt & Yang, 2004), **Amazon** product reviews from He & McAuley (2016), and finally clinical notes from **MIMIC-III** (Johnson et al., 2016) with personal identifiable information (PII) masked out. Our choice of domains highlights legal risks discussed in the earlier sections, e.g., CC-News, BookCorpus2, Books3 and Amazon reviews are mostly copyrighted, Github is mostly not permissively licensed,[10] and Enron Emails and MIMIC-III include private text.

---

[10]Kocetkov et al. (2023) estimates about 13% of the Github data is under MIT, Apache, and BSD, permissive licenses defined by Together (2023).

Table 11: Perplexity on the parametric LMs trained on $\overline{\text{PD}}$, $\overline{\text{PDSW}}$, and $\overline{\text{PDSWBY}}$, as well as Pythia 1.4B, a model trained with similar amounts of compute but on non-permissive data. We use ■, ■, and ■ to indicate text that is in-domain, out-of-domain, or out-of-domain but has relevant data in-domain data (e.g., non-permissive Github code versus our permissive training code). Reported on the validation data; see Table 2 for results on the test data.

| Eval data | $\overline{\text{PD}}$ | $\overline{\text{PDSW}}$ | $\overline{\text{PDSWBY}}$ | Pythia |
|---|---|---|---|---|
| FreeLaw | 5.3 | 5.7 | 6.5 | 5.6 |
| Gutenberg | 14.6 | 11.9 | 13.4 | 12.7 |
| HackerNews | 36.6 | 12.1 | 13.2 | 12.5 |
| Github | 13.3 | 2.6 | 2.7 | 2.4 |
| NIH ExPorter | 28.6 | 19.3 | 15.1 | 11.2 |
| PhilPapers | 55.2 | 24.2 | 16.5 | 14.3 |
| Wikipedia | 27.9 | 19.7 | 11.1 | 9.0 |
| CC News | 30.8 | 21.3 | 19.3 | 10.9 |
| BookCorpus2 | 25.2 | 19.2 | 20.2 | 12.8 |
| Books3 | 25.9 | 18.7 | 18.1 | 12.4 |
| OpenWebText2 | 38.1 | 21.2 | 18.8 | 11.5 |
| Enron Emails | 19.9 | 14.3 | 14.5 | 7.6 |
| Amazon | 81.9 | 34.7 | 37.0 | 22.8 |
| MIMIC-III | 18.2 | 16.4 | 13.6 | 11.5 |
| Average | 30.1 | 17.2 | 15.7 | 11.2 |

We merge all text into one stream of text and split them into batches with a maximum sequence length of 1,024 and a sliding window of 512, a setup that is standard in prior language modeling literature (Baevski & Auli, 2019; Khandelwal et al., 2020). For MIMIC-III, which includes masked personally-identifiable information (PII), we filter out notes where more than 50% of tokens correspond to PII, and then exclude tokens corresponding to PII when computing perplexity.

### C.4 DETAILS OF DOWNSTREAM TASK EVALUATION

We perform zero-shot prompting for nine text classification datasets: AGNews (Zhang et al., 2015), Yahoo (Zhang et al., 2015), Subj (Pang & Lee, 2004), SST-2 (Socher et al., 2013), MR (Pang & Lee, 2004), Rotten Tomatoes (RT), CR (Hu & Liu, 2004), Amazon polarity (Amz, McAuley & Leskovec (2013)) and RTE (Dagan et al., 2005). The tasks range from topic classification and sentiment analysis to subjectivity classification and textual entailment.

We use the templates and label words to map the task into a sentence completion problem, following the standard from literature (Radford et al., 2019; Brown et al., 2020; Holtzman et al., 2021). These templates and label words are taken from Shi et al. (2022), reported in Table 10. For both parametric LMs and $k$NN-LM, we apply the domain-conditional PMI scoring (Holtzman et al., 2021) for determining the probability of each label. For $k$NN-LM, we follow a method from Shi et al. (2022) which employs the fuzzy verbalizers to expand the token set associated with each output label in our task. Also following Shi et al. (2022), we use IMDB (Maas et al., 2011) with 8 million tokens as an additional datastore. We perform hyperparameter search on the validation dataset of each task, considering $k \in \{128, 512, 4196, 8192\}$, $\tau \in \{1, 3, 5, 10, 40, 80\}$, and different choice of datastores. The chosen hyperparameters are reported in Table 10.

## D ADDITIONAL EXPERIMENTAL RESULTS

Table 11 reports perplexity of the parametric LMs on the validation data that is analogous to Table 2. Table 12 reports perplexity of both parametric and nonparametric LMs on the validation data that is analogous to Table 3. Findings based on the validation data and on the test data are largely consistent.

### D.1 ABLATIONS: PARAMETRIC COMPONENT (SECTION 4.1)

**Effect of upsampling low-resource data.** Since OPEN LICENSE CORPUS has an extremely skewed distribution of domains, we upsample less-representative domains during training. Table 13 (left) compares the models trained on $\overline{\text{PDSW}}$ with and without domain upweighting. In-domain datasets

Table 12: Perplexity of parametric LMs (Prm-only), $k$NN-LM and RIC-LM; ■ indicates in-domain; ■ indicates out-of-domain; ■ indicates out-of-domain but has relevant data in-domain. Reported on the validaiton data; see Table 3 for results on the test data.

| Eval data | PDSW | | | Pythia |
|---|---|---|---|---|
| | Prm-only | $k$NN-LM | RIC-LM | Prm-only |
| Github | 2.6 | 2.4 | 2.4 | 2.4 |
| NIH ExPorter | 19.3 | 14.9 | 18.5 | 11.2 |
| Wikipedia | 19.7 | 14.1 | 18.9 | 9.0 |
| CC News | 21.3 | 7.1 | 14.8 | 10.9 |
| Books3 | 18.8 | 17.3 | 18.5 | 12.5 |
| Enron Emails | 14.3 | 6.7 | 11.1 | 7.6 |
| Amazon | 34.7 | 26.2 | 33.7 | 22.8 |
| MIMIC-III | 16.3 | 7.2 | 14.1 | 11.5 |
| Average | 18.4 | 12.0 | 16.5 | 11.0 |

Table 13: **(Left)** Effect of re-weighting rare domains, comparing models trained on OLC (PDSW) with and without upsampling. **(Right)** Effect of SW data, with and without explicit source code—we train an LM with SW data but remove all of the actual source code (i.e., we leave Hacker News, Ubuntu IRC, Deepmind Math, and AMPS). Both tables report perplexity on the validation data.

| Data | PDSW w/o upsampling | PDSW w upsampling | Data | PD | PDSW w/o code | PDSW |
|---|---|---|---|---|---|---|
| FreeLaw | 4.9 | 5.7 | FreeLaw | 5.3 | 5.7 | 5.7 |
| Github | 2.4 | 2.6 | Github | 13.3 | 8.2 | 2.6 |
| NIH ExPorter | 20.0 | 19.3 | NIH ExPorter | 28.6 | 26.2 | 19.3 |
| PhilPapers | 23.9 | 24.2 | PhilPapers | 55.2 | 36.4 | 24.2 |
| Wikipedia | 19.9 | 19.7 | Wikipedia | 27.9 | 26.5 | 19.7 |
| CC News | 21.8 | 21.3 | CC News | 30.8 | 28.8 | 21.3 |
| BookCorpus2 | 19.4 | 19.2 | BookCorpus2 | 25.2 | 23.8 | 19.2 |
| OpenWebText2 | 21.0 | 21.2 | OpenWebText2 | 38.1 | 31.7 | 21.2 |
| Enron Emails | 13.5 | 14.3 | Enron Emails | 19.9 | 18.5 | 14.3 |
| Amazon | 35.7 | 34.7 | Amazon | 81.9 | 46.1 | 34.7 |

that are not upweighted, e.g., FreeLaw, see slight degration in performance. On out-of-doain datasets, there is no significant differences, although the model with upsampling is marginally better (19.6 vs. 19.7 when averaged over 9 out-of-domain datasets). We note that we did not tune the upweighting ratio nor explore alternative upweighting approaches (Xie et al., 2023) due to resource constraints, and leave them for future work.

**59B tokens of source code significantly help.** When using SW, a substantial $59.1\%$ of the training data is actual source code. To determine SW provides such large gains, we also run an ablation where we include SW data but exclude all of the actual source code, i.e., we only include Hacker News, Ubuntu IRC, Deepmind Math, and AMPS on top of the PD data. This leaves models trained on 99.6B tokens for OLC (PDSW) and 40.7B for OLC (PDSW) excluding source code. Table 13 (right) report results on a subset of the validation domains. Including source code provide significant benefits for certain test datasets, e.g., nearly a 20 point improvement in perplexity on PhilPapers, likely because it significantly increases the size of the training data.

**Performance gaps with Pythia are strongly correlated to domain shift.** We hypothesize that the performance differences we see between our models and Pythia can be explained by domain shift between our corpus and the evaluation domains. To confirm these effects, we show a scatterplot that describes the relationship between ngram overlap of the OLC and the Pile (from §B.2 with the performance gap between PDSW and Pythia in Figure 6. There is a strong negative correlation between these two metric ($r$=-0.72, $p < 0.005$).

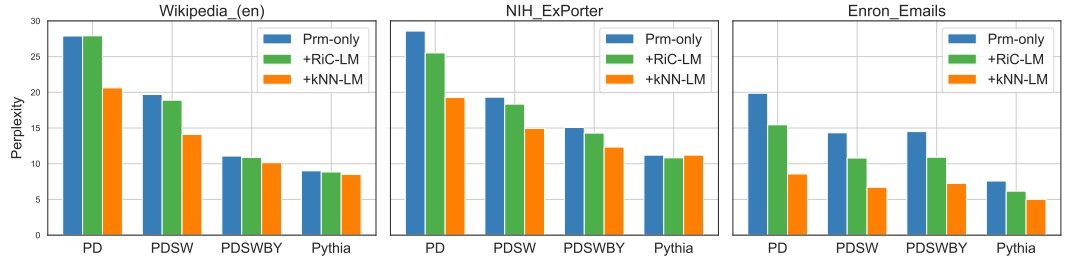

Figure 3: Results on different variants of models: $\overline{\text{PD}}$, $\overline{\text{PDSW}}$ and $\overline{\text{PDSWBY}}$ variants of SILO as well as Pythia. Adding a nonparametric component through either RIC-LM and $k$NN-LM helps and $k$NN-LM is overall better than RIC-LM, consistently across all models and evaluation datasets.

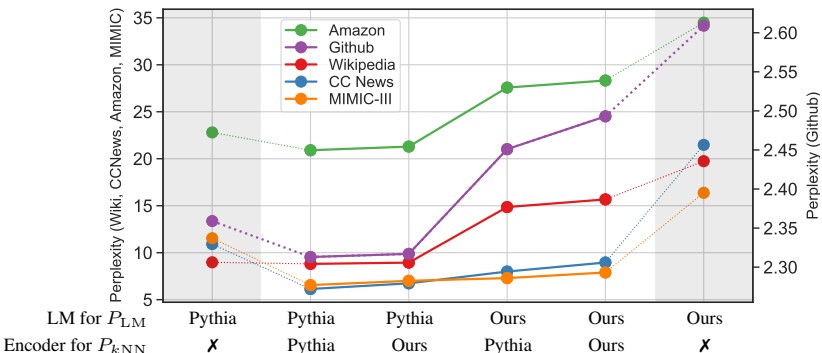

Figure 4: Impact of using different parameters on SILO. Perplexity on random 128K tokens from the validation data reported. The left-most and the right-most models are parametric models, and the other four models are $k$NN-LMs, using a datastore with 204.8 million tokens (20% of the datastore we use for the main experiments). *Ours* indicates our parametric model trained on the $\overline{\text{PDSW}}$ subset of OPEN LICENSE CORPUS. **Most of the performance degradation comes from using the out-of-domain parametric LM, rather than using the out-of-domain encoder.**

## D.2 ABLATIONS: ADDING THE NONPARAMETRIC COMPONENT (SECTION 4.2)

**Impact of adding a nonparametric component across varying LMs.** We compare parametric-only LM, RIC-LM and $k$NN-LM over four different LMs: $\overline{\text{PD}}$, $\overline{\text{PDSW}}$ and $\overline{\text{PDSWBY}}$ variants of SILO as well as Pythia. Figure 3 reports their results on three evaluation datasets: Wikipedia, NIH ExPorter and Enron Emails. Findings from Section 4.2 hold across all models: both RIC-LM and $k$NN-LM are consistently better than the parametric-only LM, and $k$NN-LM overall achieves the best performance. For instance, $k$NN-LM allows the model to be comparable to or outperform the one-level relaxed variant, e.g., a $\overline{\text{PD}}$-based $k$NN-LM is comparable to a $\overline{\text{PDSW}}$-based parametric LM, and $\overline{\text{PDSW}}$-based $k$NN-LM is comparable to a $\overline{\text{PDSWBY}}$-based parametric LM.

**Where does the remaining gap come from?** Even when scaling the datastore with $k$NN-LM, SILO lags behind Pythia on a few domains. Moreover, a Pythia-based $k$NN-LM outperforms our model since $k$NN-LM improves Pythia as well. There are two possible points of failure in our model for these cases: either the parametric component (which outputs $P_{\text{LM}}$) struggles out-of-domain, or the encoder (that outputs $P_{k\text{NN}}$) struggles out-of-domain. To better understand which part of the model contributes to the gap we observe, we vary SILO with different choices for the parametric component and the encoder. We compare replacing either the parametric component or the encoder with Pythia. This setup allows us to measure the effects of the out-of-domain nature of our parametric component (which is only trained on $\overline{\text{PDSW}}$ subset of OLC) in each of these components.

Results in Figure 4 reveal that most performance gaps come from the LM: performance improves significantly when the parametric component is replaced with Pythia, given a fixed encoder. In contrast, performance improvement is relatively marginal when the encoder is replaced with Pythia,

Table 14: Ablations on different variants of RIC-LMs. Perplexity on the validation data reported. *ensbl-10* is 10x slower than other three methods.

| Data | PDSW | | | | Pythia | | | |
|---|---|---|---|---|---|---|---|---|
| | basic | ensbl-10 | concat-2 | concat-next | basic | ensbl-10 | concat-2 | concat-next |
| CC News | 14.8 | 13.5 | 17.0 | 18.8 | 8.2 | 7.9 | 9.2 | 9.9 |
| Enron Emails | 11.1 | 10.0 | 12.8 | 13.4 | 6.3 | 6.117 | 7.1 | 7.3 |

given a fixed parametric component. These results indicate that the parametric component, which gives $P_{\text{LM}}$, is quite sensitive to domain shift, but the encoder, which provides the nonparametric distribution $P_{k\text{NN}}$, is fairly robust to extreme domain shift. This also explains why $k$NN-LM generalizes better than RIC-LM, since RIC-LM is bottlenecked by the parametric component.

**Effect of scaling the datastore in-domain and out-of-domain.** §4.2 shows that performance of both $k$NN-LM and RIC-LM rapidly improves as the datastore size grows, and $k$NN-LM improves more rapidly than RIC-LM does. This evaluation is mainly done with SILO where the test domains are out-of-domain. Does this trend hold when the test domains are in-domain? To answer this question, we examine effect of scaling the datastore with Pythia 1.4B, where all of our test datasets can be considered in-domain.

Figure 5 reports the results: Pythia on the left, SILO (PDSW) on the right. Results show that both Pythia and SILO see consistent improvements from $k$NN-LM and RIC-LM as the datastore gets larger, although the slope is larger with SILO than with Pythia. Again consistent to findings in §4.2, $k$NN-LM scales better than RIC-LM does, resulting in $k$NN-LM outperforming RIC-LM with a reasonably large datastore in most cases (with an exception of Pythia on Github, where RIC-LM outperforms $k$NN-LM with a reasonable size of a datastore).

**Ablations on variants of RIC-LM.** We explore four different variants of RIC-LM. (1) The **basic** is the method described in §3.2, which uses text blocks with a length of $L$ each. At inference, it takes the top 1 text block from the datastore and feeds it to the LM, i.e., $P_{\text{LM}}(y|\hat{b}, x)$ where $x$ is the input and $\hat{b}$ is the top 1 text block. (2) The **ensbl-$k$** ($k = 10$) variants is also based on text blocks with a length of $L$ each. At inference, it takes the top $k$ text blocks from the datastore, feeds each to the LM in parallel and aggregates the probability distributions, i.e., $\frac{1}{k} \sum_{1 \le i \le k} P_{\text{LM}}(y|\hat{b}_i, x)$ where $\hat{b}_1...\hat{b}_k$ are the top $k$ text blocks. This follows a method from Shi et al. (2023). (3) The **concat-$k$** ($k = 2$) variant uses text blocks with a length of $\frac{L}{k}$ each. At inference, it takes the top $k$ text blocks from the datastore, concatenates them in a reverse order, and feeds it into the LM, e.g., $P_{\text{LM}}(y|\hat{b}_k, \cdots, \hat{b}_1, x)$ where $\hat{b}_1...\hat{b}_k$ are the top $k$ text blocks. (4) The **concat-next** variant uses text blocks with a length of $\frac{L}{2}$ each. At inference, it takes the top 1 text block from the datastore, concatenates the text block and the subsequent text block in a datastore, and feeds it into the LM. This is based on the intuition that the continuation of the text block that is most similar to the query can be useful for the continuation of the query; Borgeaud et al. (2022) has explored a similar approach based on the same intuition. We use $L = 1024$ for all variants. The ensbl-$k$ variant has run-time that is approximately $k$ times of run-time of the basic, concat-$k$ and concat-next.

Results are reported in Table 14. The concat-2 and concat-next variants perform poorly, while the ensbl-10 outperforms the basic variant. However, we reached the conclusion that the significant run-time cost (i.e., 20x compared to a parametric LM) does not justify the improvements, and thus, we primarily use the basic variant for the remaining experiments. Future work may involve re-evaluating models using the ensbl-$k$ approach or enhancing its run-time efficiency.

**Effect of different approximation methods for L2 distance.** Prior work (Khandelwal et al., 2020) typically uses approximate nearest neighbor search to find the top $k$ nearest neighbors, and then computes the exact L2 distance using the original vectors. However, this may be inefficient in disk memory usage and run-time speed, due to needing to store large, original vectors and access them on-disk. We thus explore a few alternatives: (1) quantizing the original vectors to compute the L2 distance (but less aggressively than quantization for the nearest neighbor search index, thus it provides different levels of approximations for search and for L2 distance), or (2) completely dropping the original vectors and relying on approximated L2 distance from the index with aggressive quantization.

Table 15 shows that all approximation methods only marginally affect performance. For this reason, we use the most aggressive approximation that completely drops the original embeddings at the cost of about 0.5% lose in performance while using < 2% of the memory footprint. Future work may study more accurate and efficient approximation methods.

**Runtime speed.** Table 16 presents the runtime speed of the parametric LM, RIC-LM, and $k$NN-LM on the Wikipedia validation set. Speed is reported in tokens per second with a batch size of 1 using a single NVIDIA RTX 6000 GPU.

The results show that the parametric LM is notably faster than both RIC-LM and $k$NN-LM, and RIC-LM is faster than $k$NN-LM. Speed is slower as the datastore gets larger (for both RIC-LM and $k$NN-LM) and the nearest neighbor search gets less accurate (for $k$NN-LM; indicated by the number of probe $p$). $k$NN-LM can eventually match RIC-LM's speed while surpassing its performance by using a smaller datastore and less accurate search, i.e., when using 102M tokens with $p = 1$.

However, the machine used for benchmarking has a very slow IO speed, leading to an underestimation of both RIC-LM and $k$NN-LM's runtime speed, and the comparison can significantly vary based on the hardware. Either way, $k$NN-LM is currently substantially slower than a parametric LM, leaving room for potential future improvements.

**Qualitative examples.** Figure 17 provides six qualitative examples on the top-1 context retrieved by SILO-based $k$NN-LM. The model is able to assign a high probability to the ground truth token by retrieving highly relevant context, e.g., given the context (hockey) and the first name of the player, being able to retrieve the last name of the player, given the context (a show and its host), being able to complete the quote. These examples also highlight that a nonparametric approach addresses specific legal risks that we have discussed earlier, e.g., it assigns per-token attribution for free, and can provide a copyright notice when part of copyrighted text is being used for the probability distribution.

## E  DISCUSSION & FUTURE WORK

Our work suggests that it is possible to improve the trade-off between legal risk and model performance when training LMs. Our approach provides new options for model designers to mitigate the legal risk of LMs, and empowers stakeholders to have more control over the data that drives these systems. We point out a number of rich areas for future work, beyond what was mentioned throughout the paper:

**Addressing the limitations of SILO.** SILO does not completely eliminate legal risk. Instead, it provides users more control over the model's generated content and functionalities to better align with legal regulations. For instance, SILO does not remove the need for obtaining permission to use copyrighted content in a datastore when providing attribution is not sufficient, but its opt-out capabilities can strengthen fair use defense. Moreover, SILO does not prevent copying copyright content from a datastore, but it offers a way to prevent generating sensitive text (Huang et al., 2023a) or prevent copying the content verbatim. These functionalities increase the likelihood of a successful fair use defense if used appropriately.

Furthermore, while SILO mitigates copyright and privacy risks, it may exacerbate certain fairness issues, like toxicity towards marginalized groups and racial biases, especially due to the prevalence

Table 15: Ablations on approximation methods on the validation data of Wikipedia, using the LM trained on PDSW and the datastore consisting of 51.2 million tokens (5% of the datastore in the main experiments). Relative disk memory usage reported (considering *no approximation* as 1.0).

| Method | PPL | Disk use |
|---|---|---|
| Param-only | 19.7 | 0.0 |
| No approximation | 16.4 | 1.0 |
| Quantized (4x) | 16.6 | 0.25 |
| Quantized (8x) | 16.6 | 0.125 |
| Quantized (16x) | 16.8 | 0.0625 |
| IVFPQ approximation | 16.8 | 0.0178 |

Table 16: Comparison in runtime speed (# tokens per second) on the validation data of Wikipedia. $p$ indicates the number of probe, one of the hyperparameters in fast nearest neighbor search ($p = 8$ in all experiments in the paper if not specified otherwise).

| Method | PPL | # tokens/s |
|---|---|---|
| Param-only | 19.7 | 1828.6 |
| RIC-LM (51.2M) | 19.3 | 812.7 |
| RIC-LM (102.4M) | 19.2 | 731.4 |
| RIC-LM (204.8M) | 19.1 | 588.5 |
| RIC-LM (409.6M) | 18.9 | 478.5 |
| RIC-LM (1,178M) | 18.9 | 419.7 |
| $k$NN-LM (51.2M) | 16.8 | 184.2 |
| $k$NN-LM (102.4M) | 16.3 | 112.0 |
| $k$NN-LM (204.8M) | 15.7 | 59.3 |
| $k$NN-LM (409.6M) | 15.0 | 31.8 |
| $k$NN-LM (1,024M) | 14.2 | 14.2 |
| $k$NN-LM (102M, $p = 1$) | 16.7 | 560.8 |
| $k$NN-LM (1,024M, $p = 1$) | 14.6 | 71.1 |
| $k$NN-LM (1,024M, $p = 2$) | 14.4 | 45.5 |
| $k$NN-LM (1,024M, $p = 4$) | 14.2 | 27.0 |

of older copyright-expired books in the training data. Exploring the balance between legal risk mitigation and fairness is an important future direction.

Finally, our study relies on explicit metadata to identify licenses, which may lead to underestimates of the amount and diversity of permissively licensed text actually available on the web. Future research may investigate *inferring* data licenses from documents in web crawl at scale, which may be an effective way to build more heterogeneous, permissively licensed corpora.

**Introducing novel data licensing approaches.**    SILO introduces the possibility for data owners to set different levels of permissivity for learning parameters and for including in a nonparametric datastore. A data owner might choose to be more permissive about including data in the datastore due to its ease of removal, ensuring that the excluded data has no influence on model predictions anymore, and its ability to provide per-prediction attribution. Moreover, we envision that SILO could provide a path forward for data owners to get properly credited (or be paid directly) every time their data in a datastore contributes to a prediction. This is orthogonal to recent work that circumvented copyright issues by licensing out training data from data creators (Yu et al., 2023).

**Investigating other copyright risk mitigation strategies.**    It is critical to continue to develop new techniques that use copyrighted data while protecting the rights of data owners and subjects. In addition to nonparametric approaches, there are many other ways to achieve these goals. First, one could train LMs on copyrighted content but filter and guide their outputs towards text that is non-infringing (Henderson et al., 2023). Second, training models with differential privacy (Dwork et al., 2006; Abadi et al., 2016) or near-access freeness (Vyas et al., 2023) may prevent them from regenerating individual details of copyright data. Finally, one could provide attributions for standard base LMs using post-hoc attribution methods, e.g., influence functions (Koh & Liang, 2017), rather than switching the model class to a retrieval-based model. All of these methods are complementary and orthogonal to our proposed approach.

**Generalizing SILO as a modular language model.**    Our work is closely related to recent studies on modular LMs, which have specialized parameters (or *experts*) trained on different domains (Gururangan et al., 2022; Li et al., 2022; Gururangan et al., 2023), languages (Pfeiffer et al., 2020; 2022), or tasks (Chen et al., 2022b; Jang et al., 2023a). Our work extends modular LMs to include nonparametric datastores, and focuses on *specializing* different parts of the model to low- and high-risk subsets of the training data. Legal risks may also be mitigated with a collection of parametric expert models that are specialized to low- and high-risk data. Future work may explore this possibility as well as the usefulness of combining a nonparametric datastore with parametric experts.

**Extending SILO to other modalities.**    While this work focuses on text-only models, similar methods to ours could apply to other domains and modalities. For instance, it might be possible to build permissive text-to-image generative models (Rombach et al., 2022) using compartmentalized public domain pre-training and retrieval-augmentation (Chen et al., 2022a; Golatkar et al., 2023). We believe such approaches are especially promising because there are many sources of public domain data in other modalities, e.g., images, speech, video, and more.

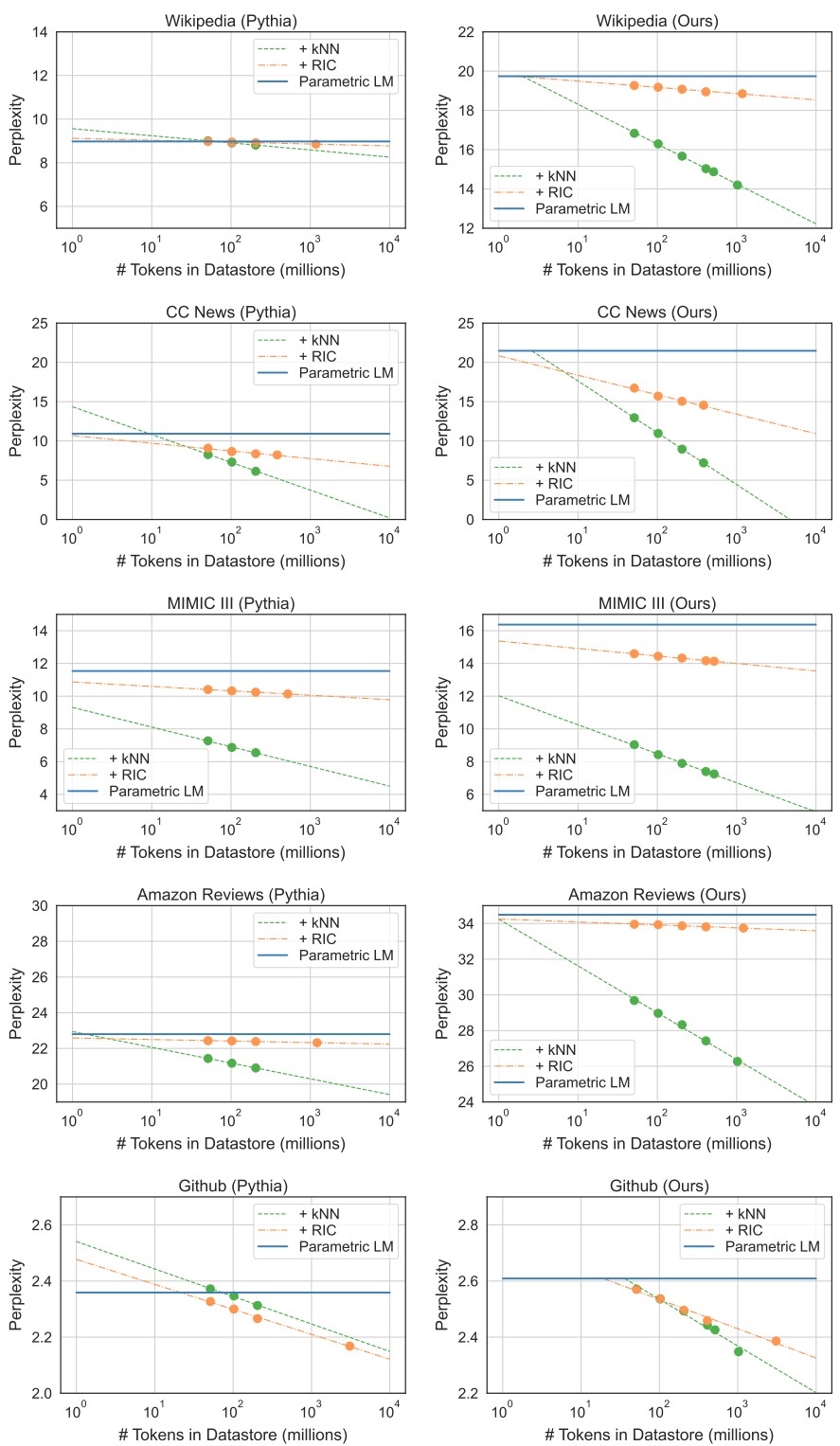

Figure 5: Comparison between parametric LM, RIC-LM and $k$NN-LM on five domains, with Pythia (left) and SILO PDSW (right), respectively. Perplexity on random 128K tokens from the validation data reported.

Table 17: Qualitative examples of retrieved context of our model. **Red underlined text** indicates the next token that immediately follows the prefix. The first two are from Github; the next two are from Enron Emails; and the last two are from CC News.

**Test Prefix**
include '../lib/admin.defines.php';
include '../lib/admin.module.access.php';
include '../lib/admin.smarty.php';
if (! has_rights (
**Test Continuation** ACX_BILLING)) { Header . . .
**Retrieved Prefix**
(...)
* You should have received a copy of the GNU Affero General Public License
* along with this program. If not, see <http://www.gnu.org/licenses/>.
*
*
**/
if (! has_rights (
**Retrieved Continuation** ACX_ACCESS)) { Header ...

---

**Test Prefix**
0x5f #define S5K4AA_DEFAULT_BRIGHTNESS 0x10
/*****************/
/* Kernel
**Test Continuation** module parameters */ extern int force_sensor; ...
**Retrieved Prefix**
* Copyright © 2011-2013 Jozsef Kadlecsik <kadlec@blackhold.kfki.hu>
*
* This program is free software; you can redistribute it and/or modify
* it under the terms of the GNU General Public License version 2 as
* published by the Free Software Foundation.
*/
/* Kernel
**Retrieved Continuation** module implementing an IP set type: . . .

---

**Test Prefix** . . . Mark or credit about hedge funds? Sara
Sara Shackleton
Enron North America Corp.
[Address]
[Phone number]
[Email address]
— Forwarded by Sara Shacleton/HOU/ECT on 01/2023/2022 05:41PM —
Tana
**Test Continuation** Jones    12/14/2000
**Retrieved Prefix** ... Food will be provided! Tana: Please feel free to extend the invitation to any Enron employees who may be interested in te presentation. 1st come, 1st serve. Thanks, Sylvia. — Forwarded by Sylvia Hu/Corp/Enron on 07/14/2000 03:17PM — Tana
**Retrieved Continuation** Jones@ECT. 07/13/2000

---

**Test Prefix** Ken Lay and Jeff Skilling were interviewed on CNNfn to discuss the succession of Jeff to CEO of Enron. (...) and then choose "Enron's Succession Plan.". The interview will be available every 15 minutes
**Test Continuation** through Friday, Dec. 15.
**Retrieved Prefix** Did you miss Jeff on CNBC "Street Signs" yesterday? Not to worry. (...) and then choose > "Skilling CNBC.". The interview will be available every ten minutes
**Retrieved Continuation** through > Wednesday, Dec. 6.

---

**Test Prefix** . . . The teams toured the city, explored west Edmonton mall and also got to take in an Oilers practice where they met German hockey star Leon
**Test Continuation** Draisaitl
**Retrieved Prefix** One minute and 19 seconds later, Cannor McDavid took a pass from Leon
**Retrieved Continuation** Draisaitl

---

**Test Prefix** ... Foley on RAW's run-time issues. Claiming that having the show run so late is one of the reasons why the final hour of RAW tends to struggle, Foley didn't end there. "No one else at 10:30pm is a
**Test Continuation** PG show. I won't say that across
**Retrieved Prefix** . . . way to the ring' podcast Foley cited RAW's duration and RG rating as hindrances to the show's popularity. Here's what he had to say: "Sometiems we try to look into the reasons why the third hour doesn't perform as well as the first two, and I'm like 'well that's because people go to bed! No one else at 10:30pm is a
**Retrieved Continuation** PG show. I won't say that across

Table 18: Unigram and bigram overlap between the domain of the Pile validation data and the domains of OPEN LICENSE CORPUS.

| Dataset | BookCorpus2 | Books3 | Enron Emails | FreeLaw | Github | Gutenberg (PG-19) |
|---|---|---|---|---|---|---|
| ccby_law | 0.02 | 0.04 | 0.02 | 0.08 | 0.02 | 0.03 |
| ccby_s2orc | 0.05 | 0.10 | 0.03 | 0.06 | 0.05 | 0.07 |
| ccby_stackexchange | 0.05 | 0.07 | 0.03 | 0.04 | 0.16 | 0.05 |
| ccby_stackoverflow | 0.03 | 0.05 | 0.01 | 0.03 | 0.07 | 0.03 |
| ccby_wikinews | 0.07 | 0.15 | 0.02 | 0.08 | 0.03 | 0.09 |
| ccby_wikipedia | 0.05 | 0.11 | 0.02 | 0.06 | 0.03 | 0.08 |
| pd_books | 0.13 | 0.26 | 0.03 | 0.07 | 0.04 | 0.33 |
| pd_law | 0.05 | 0.10 | 0.02 | 0.35 | 0.03 | 0.07 |
| pd_news | 0.06 | 0.14 | 0.02 | 0.07 | 0.02 | 0.08 |
| pd_s2orc | 0.08 | 0.15 | 0.04 | 0.08 | 0.05 | 0.13 |
| sw_amps_math | 0.01 | 0.02 | 0.01 | 0.01 | 0.02 | 0.01 |
| sw_dm_math | 0.00 | 0.01 | 0.00 | 0.01 | 0.01 | 0.01 |
| sw_github | 0.04 | 0.04 | 0.03 | 0.03 | 0.24 | 0.04 |
| sw_hackernews | 0.09 | 0.18 | 0.03 | 0.06 | 0.06 | 0.10 |
| sw_ubuntu_irc | 0.12 | 0.11 | 0.06 | 0.04 | 0.08 | 0.09 |

| Dataset | OpenWebText2 | PhilPapers | Wikipedia (en) | cc-news | new-amazon | HackerNews |
|---|---|---|---|---|---|---|
| ccby_law | 0.02 | 0.03 | 0.05 | 0.05 | 0.02 | 0.03 |
| ccby_s2orc | 0.05 | 0.11 | 0.02 | 0.06 | 0.06 | 0.07 |
| ccby_stackexchange | 0.06 | 0.08 | 0.04 | 0.05 | 0.06 | 0.10 |
| ccby_stackoverflow | 0.05 | 0.07 | 0.02 | 0.03 | 0.07 | 0.06 |
| ccby_wikinews | 0.09 | 0.22 | 0.02 | 0.04 | 0.09 | 0.08 |
| ccby_wikipedia | 0.06 | 0.17 | 0.02 | 0.07 | 0.07 | 0.06 |
| pd_books | 0.16 | 0.15 | 0.03 | 0.11 | 0.11 | 0.09 |
| pd_law | 0.06 | 0.10 | 0.02 | 0.06 | 0.06 | 0.05 |
| pd_news | 0.08 | 0.21 | 0.02 | 0.06 | 0.09 | 0.06 |
| pd_s2orc | 0.08 | 0.13 | 0.03 | 0.09 | 0.08 | 0.09 |
| sw_amps_math | 0.01 | 0.02 | 0.01 | 0.01 | 0.01 | 0.02 |
| sw_dm_math | 0.00 | 0.01 | 0.00 | 0.00 | 0.00 | 0.00 |
| sw_github | 0.04 | 0.04 | 0.03 | 0.05 | 0.03 | 0.06 |
| sw_hackernews | 0.14 | 0.20 | 0.04 | 0.06 | 0.16 | 0.19 |
| sw_ubuntu_irc | 0.13 | 0.10 | 0.10 | 0.10 | 0.08 | 0.18 |

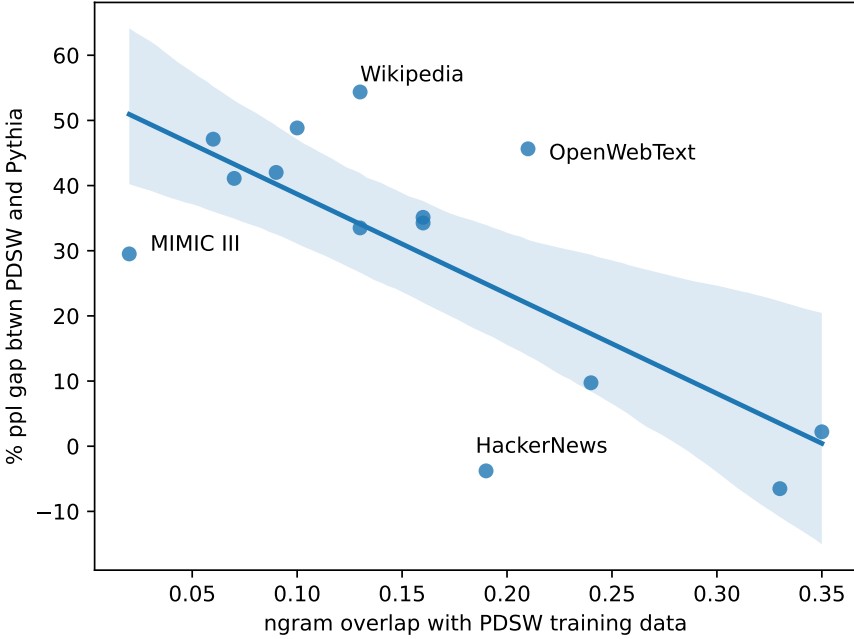

Figure 6: There is a strong negative correlation between ngram overlap of a domain with the PDSW training data and the perplexity gap between the PDSW LM and Pythia ($r$=-0.72, $p < 0.005$).

