# OpenReview forum: "SILO Language Models: Isolating Legal Risk In a Nonparametric Datastore"
_ICLR.cc/2024/Conference — ICLR 2024 spotlight_

### Official Review · Reviewer_mHsD · 2023-10-31

**Soundness:** 4 excellent
**Presentation:** 4 excellent
**Contribution:** 4 excellent
**Rating:** 8
**Confidence:** 3

**Summary:**

The methodology presented in the paper attempts to address current concerns regarding language technology, namely its reliance on copyrighted data. The authors propose a new corpus, the 228B-token Open Licence Corpus (OLC) comprising texts with various permissive licenses. Additionally, they show that since OLC is less broad in content than its alternatives that do not eschew copyrighted text, training a language model solely on this results in comparably low performance.

In order to benefit from using copyrighted texts, the authors propose to use their language model trained on OLC in conjunction with a datastore containing copyrighted text. They test two distinct retrieval methods, k-nearest neighbours (KNN) and retrieval-in-context (RIC) for returning tokens (in the case of KNN) or text blocks (in the case of RIC) from the datastore. They find that by using KNN-based retrieval they can achieve performance close to Pythia, a language model trained on non-permissive data across 14 domains on perplexity and 10 downstream tasks.

SILO models such as the one proposed by the authors are conducive to both identifying the pieces of copyrighted texts contributing to a given decision, as well as removing such texts, complying with concepts like fair use and GDPR.

**Strengths:**

Overall: The paper is fairly clearly written, it addresses a very important and current problem in the application of language technology, and carries out extensive analyses. It also compares performance of the proposed method across a large number of domains and tasks.

Soundness: There are still some costs associated with inference time and performance when using SILO models as opposed to using “classical” models trained on a range of permissive and non-permissive texts. However, the methodology proposed by the authors seems like a viable method of mitigating the very current problem of the use of copyrighted or otherwise protected corpora in language technology.

Contribution: The authors create OLC, a diverse pretraining corpus made up of permissive texts. Additionally, they train a language model on this corpus. They also present a large number of analyses untangling the impact of various factors relating to domain overlap and retrieval methods, among all.

**Weaknesses:**

Overall: Some of the tables are difficult to read, as mentioned, adding some highlights to help compare the performance would have been helpful. Additionally, in the case of Table 3, the percentages included are confusing. Are the minuses referring to the advantage (or lack thereof) when compared with Pythia? If yes, why do we get -100% when the result is the same across SILO and Pythia?

Presentation: The content is mostly presented clearly. However, it would have helped to have some examples at least in the appendix or a more descriptive figure illustrating KNN and RIC-based retrieval to make it somewhat easier to understand. Otherwise, some of the denser tables would have benefitted from adding highlights (underline or bold text) to important scores to lead the eye.

**Questions:**

I have no questions, but please make sure that you improve clarity when it comes to presenting your results. Also, I think being a bit more explicit in how retrieval works at inference time in practice could improve the paper and match its otherwise accessible tone.

---

> ### Author Response · Authors · 2023-11-17
>
> We thank the reviewer for their constructive comments/feedback and for supporting our paper. We respond to the reviewer’s comments below.
>
> > Overall: Some of the tables are difficult to read, as mentioned, adding some highlights to help compare the performance would have been helpful. Additionally, in the case of Table 3, the percentages included are confusing. Are the minuses referring to the advantage (or lack thereof) when compared with Pythia? If yes, why do we get -100% when the result is the same across SILO and Pythia?
>
> The percentages indicate how much gap between parametric-only SILO and Pythia was reduced by adding a nonparametric datastore, e.g., -100% indicates it completely removes the gap. We agree this is confusing - we will revise the table to make the results easier to interpret.
>
> > Presentation: The content is mostly presented clearly. However, it would have helped to have some examples at least in the appendix or a more descriptive figure illustrating KNN and RIC-based retrieval to make it somewhat easier to understand. Otherwise, some of the denser tables would have benefitted from adding highlights (underline or bold text) to important scores to lead the eye. Also, I think being a bit more explicit in how retrieval works at inference time in practice could improve the paper and match its otherwise accessible tone.
>
> We will include the overview figure for kNN and RIC in the extra page in the final version of the paper. Hopefully, this helps the readers to understand how retrieval works at inference time as well.

---

> > ### Comment · Reviewer_mHsD · 2023-12-04
> > **Response to author rebuttal**
> >
> > Thanks for your response to my comments.
> >
> > I still stand by my scores for this paper.

---

### Official Review · Reviewer_P6ek · 2023-11-01

**Soundness:** 3 good
**Presentation:** 3 good
**Contribution:** 3 good
**Rating:** 8
**Confidence:** 4

**Summary:**

This paper verifies the feasibility of decomposing data into high-risk and low-risk categories to ensure the legality of LLMs. Low-risk data is employed during the pretraining phase, whereas high-risk data can be integrated during inference by utilizing in-context retrieval augmentation or kNN-LM.

**Strengths:**

* This paper introduces a paradigm aimed at ensuring the legality of data used in LLMs, with the potential to significantly impact related research areas such as LLM privacy and security.
* It conducts a comparative analysis of two approaches for incorporating high-risk data into LLMs during the inference process, offering valuable insights into their effectiveness.
* The paper also introduces a dataset that can support further research in this line of investigation.

**Weaknesses:**

* This paper primarily focuses on evaluating the zero-shot performance of SILO. Given that pretrained LLMs usually serve as base models for instruction-tuning or task fine-tuning, understanding the extent of SILO's influence on these processes is of significant importance. However, this is not explored in this paper.
* In the evaluation of SILO, this paper primarily relies on PPL as a metric, although it does carry out experiments on text classification tasks. A more comprehensive assessment across various settings, a broader range of tasks, and in terms of multiple aspects (such as helpfulness and harmfulness) could offer a deeper and more insightful understanding of SILO's performance.

**Questions:**

In the paper, the authors claim that "Pythia ... is trained with a similar amount of compute (data size and model parameters)." What are the exact computational costs when training SILO and Pythia?

---

> ### Author Response · Authors · 2023-11-17
>
> We thank the reviewer for their helpful feedback and for supporting our paper. We respond to the reviewer’s comments and questions below.
>
> > This paper primarily focuses on evaluating the zero-shot performance of SILO. Given that pretrained LLMs usually serve as base models for instruction-tuning or task fine-tuning, understanding the extent of SILO's influence on these processes is of significant importance. However, this is not explored in this paper.
>
> We agree instruction-tuning and task fine-tuning will be important to explore in future work. As described in our response to reviewer cDVV, we focus on LM perplexity and zero-shot text classification in the paper due to the following two reasons:
> Prior work has shown that LM perplexity correlates well with downstream tasks across a large collection of tasks and all models [1, 2, 3].
> Prior work has shown that adding a nonparametric datastore improves a range of tasks in both fine-tuning scenarios [4, 5] and instruction tuning [6] even when pretrained LMs are already trained on in-domain data. Gains are expected to be larger in SILO where pretrained LMs are not trained on in-domain data (this is the case in perplexity, as reported in Appendix D.2 & Figure 5).
>
> * [1] Kaplan et al. "Scaling Laws for Neural Language Models". 2020. https://arxiv.org/abs/2001.08361
> * [2] Hernandez et al. Scaling Laws for Transfer. 2021. https://arxiv.org/abs/2102.01293
> * [3] Xia et al. “Training Trajectories of Language Models Across Scales”. ACL 2023. https://arxiv.org/abs/2212.09803
> * [4] Guu et al. 2020. “REALM: Retrieval-Augmented Language Model Pre-Training”. https://arxiv.org/abs/2002.08909
> * [5] Izcard et al. "Atlas: Few-shot Learning with Retrieval Augmented Language Models". 2022  https://arxiv.org/abs/2208.03299
> * [6] Lin et al. 2023. "RA-DIT: Retrieval-Augmented Dual Instruction Tuning" https://arxiv.org/abs/2310.01352
>
> > In the evaluation of SILO, this paper primarily relies on PPL as a metric, although it does carry out experiments on text classification tasks. A more comprehensive assessment across various settings, a broader range of tasks, and in terms of multiple aspects (such as helpfulness and harmfulness) could offer a deeper and more insightful understanding of SILO's performance.
>
> We agree a larger-scale evaluation in downstream tasks will be valuable. Some evaluations the reviewer suggested necessitate instruction-tuning or RLHF (e.g., helpfulness and harmfulness), which we think is out of scope for this paper but can be a good future direction.
>
> > Questions: In the paper, the authors claim that "Pythia ... is trained with a similar amount of compute (data size and model parameters)." What are the exact computational costs when training SILO and Pythia?
>
> Based on the Pythia paper [1], both Pythia and SILO are trained with A100 GPUs, each with 40GB of memory. Pythia 1.4B (the model we used for comparison as SILO has 1.3B parameters) is trained for 7,120 GPU hours. The main SILO model (PDSW) is trained for 6,613 GPU hours. We will add this information in the next version of the paper.
>
> [1] Biderman et al. "Pythia: A Suite for Analyzing Large Language Models Across Training and Scaling". 2023. https://arxiv.org/abs/2304.01373

---

### Official Review · Reviewer_X9LU · 2023-11-02

**Soundness:** 3 good
**Presentation:** 4 excellent
**Contribution:** 3 good
**Rating:** 8
**Confidence:** 4

**Summary:**

This research delves into the potential of developing technologies tailored to manage legal risks associated with the controversial issue of training Language Models (LMs) on copyrighted data. The authors introduce SILO, developed by:

1. Training a parametric LM on low-risk texts (e.g., copyright-expired books).
2. Enhancing the LM with high-risk texts (e.g., copyrighted books) stored in a nonparametric datastore, accessed only during inference.

At its core, SILO leverages the kNN-LM retrieval method (k-nearest neighbors LM by Khandelwal et al., 2020).

**Strengths:**

* The motivation behind this study is commendable, as it addresses the pressing concern over the legality of training materials for LMs.
* SILO's methodology offers an effective solution to the identified problem, as demonstrated by both its theoretical framework and experimental outcomes. Its adaptability in integrating various corpora is a significant advantage.
* Compared to alternative techniques, like post-hoc training data attribution or attempts to neutralize specific training samples' effects, this method provides robust assurances and scalability.

**Weaknesses:**

* Technological innovation is a slight drawback since the study primarily builds upon the kNN-LM framework by Khandelwal et al., 2020. However, this is somewhat mitigated by the strategic application of the technology to a pertinent issue, backed by thorough analysis and experimentation.

* As per Table 16, the runtime speed remains a notable challenge for kNN-LM. Future work should explore innovative technological solutions to address this limitation.

**Questions:**

Please briefly respond to the two weaknesses I listed.

---

> ### Author Response · Authors · 2023-11-17
>
> We thank the reviewer for their helpful feedback and for supporting our paper. We respond to the reviewer’s comments below.
>
> > Technological innovation is a slight drawback since the study primarily builds upon the kNN-LM framework by Khandelwal et al., 2020. However, this is somewhat mitigated by the strategic application of the technology to a pertinent issue, backed by thorough analysis and experimentation.
>
> As the reviewer mentioned, this work is among the first that provide technical mitigation of legal risk in LMs. The training recipe – or more broadly, the idea of separating the data for training and the data for a datastore in general – is novel and has not been explored in prior work, to the best of our knowledge. We believe the novelty of our paper lies in how the idea of separating the data can be used to mitigate legal risk–an area whose technical solutions have largely been underexplored, and empirical findings on how our model allows recovering performance close to the existing LMs. Future work may investigate new model architectures that further improve the performance and efficiency of our framework.
>
> > As per Table 16, the runtime speed remains a notable challenge for kNN-LM. Future work should explore innovative technological solutions to address this limitation.
>
> We agree – generally, we think of SILO as a way to control tradeoffs between legal risk, runtime and performance.  We believe the connection between the legal risk of non-permissive training data and a nonparametric datastore, now drawn for the first time in our work, will motivate continued research on more efficient inference with such architectures.  We will include more discussion of runtime in the extra page in the final version of the paper, and move some relevant results from Appendix D.2 to the main paper.

---

> ### Comment · Reviewer_X9LU · 2023-11-23
> **remain the rating**
>
> I have read the comments by the authors and remain with the rating.

---

### Official Review · Reviewer_cDVV · 2023-11-03

**Soundness:** 3 good
**Presentation:** 3 good
**Contribution:** 2 fair
**Rating:** 5
**Confidence:** 4

**Summary:**

The premise of this study is that some data that is legally challenging to be included during pretraining of a LLM. In this study the authors investigate whether such data can instead be used just as augmentation during inference while the LLM is only trained on uncritical data. The augmentation during inference is either achieved with the knn-LM approach or by simply adding BM25 retrieved text in the context. They investigate the effectiveness of this approach on a few zero-shot text classification benchmarks, and by measuring perplexity on text sources of various domains. They can show that a model only trained on legally safe data performs worse and adding the augmentation during inference can reduce the gap to a model that has seen such data during pretraining.

**Strengths:**

The premise hinges on the currently hotly debated issue of legal implications of using legally critical data for pretraining, and assumes that using such data in pretraining would be undesirable. They investigate the option to make it easy to remove critical data unambiguously from the inference of the model, which could enable using such data until it has to be removed.

The general quality and clarity of the paper is good, i.e. the presentation is well thought out and good to follow.

**Weaknesses:**

The general idea the paper presents is straight-forward, relevant and interesting, yet I am not convinced if it asks the right questions and measures the right things.

When data is taken away during pretraining and is only added during inference, then the first question I have is on which tasks does this matter and has a larger impact and on which tasks does it not matter. The suite of text-classification benchmarks are too simplistic to give us any relevant insights into the effect this might have.

The perplexity analysis is not too insightful, as we won't know how that will influence any of the benchmarks that might be relevant to a particular use-case. An increase in perplexity is expected under domain-shift and also that adding the inference augmentation will reduce it, so its not clear what the take-away for the reader should be (its good to have that in the appendix, but we know that knn-LM and RIC do work).

As a strong paper, I would have liked to see the potential advantages to be contrasted more with the current challenges, i.e. the paper leaves the run-time and various adjustments to make the approach feasible in the appendix, and focuses the main paper on selling the approach as a potential solution, yet realistically its far away from that.

**Questions:**

- Even with using open licence data, isn't there is still a risk, that IP from others might be infringed?
- It didn't become clear to me why the paper focuses on the different licences PD,SW and BY, as it doesn't have any significance in the experiments?

---

> ### Author Response · Authors · 2023-11-17
>
> (1/2) We thank the reviewer for their constructive comments/feedback. We respond to the reviewer’ comments and questions below.
>
> > When data is taken away during pretraining and is only added during inference, then the first question I have is on which tasks does this matter and has a larger impact and on which tasks does it not matter. The suite of text-classification benchmarks are too simplistic to give us any relevant insights into the effect this might have. The perplexity analysis is not too insightful, as we won't know how that will influence any of the benchmarks that might be relevant to a particular use-case. (...)
>
> We have evaluated our models on zero-shot classification tasks and perplexity. We agree that future work should evaluate our approach on a larger collection of tasks and see how consistent our findings are. Our focus on LM perplexity in the paper was based on the following two reasons:
> 1. Prior work has shown that LM perplexity correlates well with downstream tasks (including zero-shot and few-shot in-context learning) across a large collection of tasks and all models [1, 2, 3].  So it’s a natural first step.
> 2. Prior work has shown adding a nonparametric datastore improves performance on downstream tasks [4, 5, 6] even when pretrained LMs are already trained on in-domain data. Gains are expected to be larger in SILO where pretrained LMs are not trained on in-domain data (this is the case in perplexity, as reported in Appendix D.2 & Figure 5).
>
> Our text classification experiments confirm these findings.
>
> Furthermore, perplexity has been the primary evaluation criterion in many recent ICLR papers that innovated on language modeling [7, 8, 9, 10]. Nevertheless, we agree that conducting further downstream evaluations (especially with other classes  of models) is an exciting area for future work.
>
> * [1] Kaplan et al. "Scaling Laws for Neural Language Models". 2020. https://arxiv.org/abs/2001.08361
> * [2] Hernandez et al. Scaling Laws for Transfer. 2021. https://arxiv.org/abs/2102.01293
> * [3] Xia et al. “Training Trajectories of Language Models Across Scales”. ACL 2023. https://arxiv.org/abs/2212.09803
> * [4] Asai et al. “ACL 2023 Tutorial: Retrieval-based Language Models and Applications”. https://acl2023-retrieval-lm.github.io/
> * [5] Khandelwal et al. "Nearest Neighbor Machine Translation". ICLR 2021. https://arxiv.org/abs/2010.00710
> * [6] Jiang et al. "Active Retrieval Augmented Generation" EMNLP 2023. https://arxiv.org/abs/2305.06983
> * [7] https://openreview.net/forum?id=HklBjCEKvH ICLR 2020
> * [8] https://openreview.net/forum?id=nnU3IUMJmN ICLR 2022
> * [9] https://openreview.net/forum?id=R8sQPpGCv0 ICLR 2022
> * [10] https://openreview.net/forum?id=BS49l-B5Bql ICLR 2022
>
> > As a strong paper, I would have liked to see the potential advantages to be contrasted more with the current challenges, i.e. the paper leaves the run-time and various adjustments to make the approach feasible in the appendix, and focuses the main paper on selling the approach as a potential solution, yet realistically its far away from that.
>
> Yes, in general, we think of SILO as a way to control tradeoffs between legal risk, runtime and performance.  We believe the connection between the legal risk of non-permissive training data and a nonparametric datastore, now drawn for the first time in our work, will motivate continued research on more efficient inference with such architectures.  We will include more discussion of runtime in the extra page in the final version of the paper, and move some relevant results from Appendix D.2 to the main paper.
>
> > Even with using open licence data, isn't there is still a risk, that IP from others might be infringed?
>
> It would be great if the reviewer could elaborate what “IP from others might be infringed” means, but let us give the explanation. Generally, we cannot say SILO completely removes legal risk – legal is highly context-dependent, and there are many uncertainties about how courts will interpret copyright protections. For instance, SILO does not remove the need for obtaining permission to use copyrighted content in a datastore, but its opt-out capabilities can strengthen fair use defense. Also, SILO does not prevent copying content from a datastore, but its attribution features can make it easier to identify it. We think SILO should be interpreted as lowering legal risk and giving more control to model developers and users in trade-offs between risk and performance, rather than completely removing legal risk – as we discussed throughout the paper (Section 1 & 2 and Appendix A & E).

---

> > ### Author Response · Authors · 2023-11-17
> >
> > (2/2)
> >
> > > It didn't become clear to me why the paper focuses on the different licences PD,SW and BY, as it doesn't have any significance in the experiments?
> >
> > There are two reasons:
> > 1. We think OLC, the dataset we introduce, should inherently have varying levels of permissiveness, as what constitutes permissive or not is not a binary decision but rather a spectrum. We want to allow LM developers to draw their own boundaries and control the trade-off between the size of the training data, legal risk, and performance.
> > 2. We demonstrated that the general empirical findings hold across different possible boundaries (Appendix D.2 & Figure 3).
> >
> > We have revised the paper to make these justifications clearer.

---

### Meta-Review · Area_Chair_CS3m · 2023-12-05

**Metareview:**

This paper presents an investigation of what would happen if only low-risk texts (such as non-copyrighted and government docs) are used in large language model (LLM) training. The work uses zero-shot text classification benchmarks to compare the resulting models. The low-risk approach results in degradation in LLM performance. Authors propose mitigating this degradation by retrieving the high-risk text during inference time and using them as part of the input context, and demonstrate that with this method, it is possible to obtain higher quality LLMs while mitigating the risk from using the high-risk text during training. The proposed method is not novel, however, the investigation presented in the paper is interesting and useful and will encourage future LLM training to stick to the lower-risk resources.

**Justification For Why Not Higher Score:**

No novel method is presented, this is mainly an investigation paper.

**Justification For Why Not Lower Score:**

The findings and suggestions of the work is useful for future work!

---

### Decision · Program_Chairs · 2024-01-16

Accept (spotlight)